# Boosting Spectral Clustering on Incomplete Data via Kernel Correction and Affinity Learning

**Fangchen Yu**[1], **Runze Zhao**[1], **Zhan Shi**[1], **Yiwen Lu**[1]
**Jicong Fan**[1,2], **Yicheng Zeng**[2], **Jianfeng Mao**[1,2], **Wenye Li**[1,2]*
[1] The Chinese University of Hong Kong, Shenzhen, China
[2] Shenzhen Research Institute of Big Data, Shenzhen, China
{fangchenyu, runzezhao, zhanshi1, yiwenlu1}@link.cuhk.edu.cn
fanjicong@cuhk.edu.cn, statzyc@sribd.cn, jfmao@cuhk.edu.cn, wyli@cuhk.edu.cn

## Abstract

Spectral clustering has gained popularity for clustering non-convex data due to its simplicity and effectiveness. It is essential to construct a similarity graph using a high-quality affinity measure that models the local neighborhood relations among the data samples. However, incomplete data can lead to inaccurate affinity measures, resulting in degraded clustering performance. To address these issues, we propose an imputation-free framework with two novel approaches to improve spectral clustering on incomplete data. Firstly, we introduce a new kernel correction method that enhances the quality of the kernel matrix estimated on incomplete data with a theoretical guarantee, benefiting classical spectral clustering on pre-defined kernels. Secondly, we develop a series of affinity learning methods that equip the self-expressive framework with $\ell_p$-norm to construct an intrinsic affinity matrix with an adaptive extension. Our methods outperform existing data imputation and distance calibration techniques on benchmark datasets, offering a promising solution to spectral clustering on incomplete data in various real-world applications.

## 1 Introduction

Spectral clustering [1, 2, 3] has become a widely used and effective method for clustering non-convex data and finds diverse applications in computer vision [4, 5], natural language processing [6, 7], and bioinformatics [8, 9]. Generally, the first step in spectral clustering involves constructing an affinity matrix that captures the similarity between data points, followed by performing normalized cut [4] on the corresponding graph to partition the data into clusters. The quality of the affinity matrix is a critical factor that determines the effectiveness of the clustering performance [2, 3]. However, incomplete data is commonly seen in practice, leading to inaccurate affinities and degraded clustering performance [10, 11, 12]. As such, obtaining a high-quality affinity matrix with missing data is a challenging task that requires specialized techniques. In this paper, we aim to improve the quality of affinity matrices, which naturally enhance the performance of spectral clustering on incomplete data.

In recent years, the development of methods to construct affinity matrices with full information for spectral clustering has garnered significant attention. Two types of affinity matrices are typically utilized, namely pre-defined similarity matrices [1, 2, 13, 14] and self-expressive affinity matrices [3, 15, 16, 17, 18, 19, 20]. Pre-defined similarity matrices are easy to compute and encompass well-known kernels, including the Gaussian kernel, Exponential kernel, and Polynomial kernel. The Gaussian kernel is the most widely used due to its simplicity and effectiveness [2]. Differently, self-expressive affinity matrices learn the affinity matrix $C$ by representing each data point as a

---

*Corresponding author.

linear combination of other data points, i.e., $\min_C \frac{1}{2}\|X - XC\|_F^2 + \lambda\mathcal{R}(C)$, where the columns of $X \in \mathbb{R}^{d \times n}$ are data points, $\mathcal{R}(C)$ denotes a regularization term, and $\lambda$ is a hyperparameter, as exemplified by Sparse Subspace Clustering (SSC) [15], Low-Rank Representation (LRR) [16], and Least-Squares Representation (LSR) [17, 18]. Moreover, self-expressive methods have been expanded by integrating kernel functions into the framework, such as kernel SSC (KSSC) [19], kernel LRR (KLRR) [20], and kernel LSR (KLSR) [3], through $\min_C \frac{1}{2}\|\phi(X) - \phi(X)C\|_F^2 + \lambda\mathcal{R}(C)$, where $\phi(X)$ is a mapping function and $K := \phi(X)^\top \phi(X)$ denotes a kernel matrix. Despite the varying techniques on affinity matrices, the kernel matrix remains a cornerstone of spectral clustering for pre-defined similarities and self-expressive affinities. Nevertheless, constructing a high-quality kernel/affinity matrix is an ongoing challenge when facing incomplete data. The primary obstacle is the inability to directly calculate a kernel matrix or learn an affinity matrix due to missing values.

To address the challenge of incomplete data, data imputation techniques [21, 22, 23] such as statistical imputation and matrix completion, are conventional approaches to fill in the missing entries of the data matrix before clustering. Statistical imputation methods [24, 25], such as zero, mean, and $k$-nearest neighbors ($k$NN) imputation [26], are fast and easy to implement and can flexibly handle missing data. However, these methods may introduce bias and fail to capture the true underlying data structure, hindering the accurate affinity measures. Matrix completion methods [27, 28, 29, 30], on the other hand, aim to accurately recover the underlying structure of the data by finding a low-rank or high-rank approximation of the data matrix. Although they can handle missing data in a principled way, their performance heavily depends on the consistency of data structure and the assumptions being made, easily affected by the data distribution. Most importantly, almost all imputation methods do not offer a guarantee for the quality of the affinity matrix calculated on the imputed data [31].

Distance calibration techniques [32, 33] have emerged as an alternative solution by obtaining a high-quality distance matrix for incomplete data. These techniques calibrate an initial non-metric distance matrix estimated on incomplete data to a distance metric. Several methods have been proposed, such as the metric nearness model [34, 35, 36], which finds the nearest approximation that satisfies all triangle inequalities [37]. The double-centering algorithm [38] converts the initial non-metric distance matrix into a Euclidean distance matrix, while the Euclidean embedding method [39, 40, 41] ensures the Euclidean embeddable property is satisfied through convex optimization. Although some of these methods guarantee the quality of the calibrated distance [39], they only apply to distance-based affinities like the Gaussian kernel. Moreover, these methods may not generate high-quality affinity matrices based on calibrated distance, making it necessary to develop dedicated methods that can apply to a family of kernels and affinities with theoretical guarantees.

To address the above issues, we propose an imputation-free framework to directly learn high-quality affinity matrices via two main techniques, i.e., kernel correction and affinity learning, aiming to improve the spectral clustering performance on incomplete data, with contributions as follows.

- We propose an imputation-free framework based on kernel correction and affinity learning with improved clustering performance, providing convenient tools to deal with incomplete data. To our best knowledge, this is the first systematical work to discuss spectral clustering on incomplete data, which is a commonly seen problem in practice.

- We introduce a novel kernel correction method that directly focuses on the kernel matrix estimated from the incomplete data, and corrects it to satisfy specific mathematical properties such as positive semi-definiteness (PSD). We show that the corrected kernel matrix becomes closer to the unknown ground-truth with a theoretical guarantee, which is beneficial to spectral clustering algorithms and cannot be assured by imputation and calibration methods.

- We develop a series of new affinity learning methods to equip the self-expressive framework with the $\ell_p$-norm to capture the underlying structure of data samples better. Additionally, we combine kernel correction and affinity learning to arrive at an adaptive learning method that simultaneously learns the high-quality kernel and the self-expressive affinity matrix.

- We conduct extensive experiments that demonstrate the effectiveness of proposed methods on various benchmark datasets, showing superior results in terms of kernel estimation, spectral clustering, and affinity learning on incomplete data, compared to existing data imputation and distance calibration approaches.

## 2 Related Work

### 2.1 Spectral Clustering and Affinity Learning

**Standard Spectral Clustering** involves three steps [2]: first, a Gaussian kernel matrix $K \in \mathbb{R}^{n \times n}$ is calculated to measure the similarity between $n$ data points with $K_{ij} = \exp(-\|x_i - x_j\|^2/\sigma^2)$ where $\sigma$ is a hyperparameter. Then, an affinity graph $A \in \mathbb{R}^{n \times n}$ is constructed using the kernel matrix, which can take the form of an $\epsilon$-neighborhood graph, a fully connected graph, or a $k$-nearest neighbors ($k$NN) graph. An $\epsilon$-neighborhood graph connects pairwise points with a threshold value $\epsilon$, while a fully connected graph connects all points. Empirically, a $k$NN graph is the most popular one that connects each point to its $k$-nearest neighbors, resulting in sparse local relationships and relatively high clustering accuracy [3]. Finally, the normalized cut algorithm [1] is applied to the affinity graph $A$ to partition the data into clusters based on the normalized Laplacian matrix.

**Self-expressive Affinity Learning** is a framework to learn affinity matrices by modeling the relationships between data points, i.e., $\min_C \frac{1}{2}\|X - XC\|_F^2 + \lambda\mathcal{R}(C)$, with different types of regularization terms $\mathcal{R}(C)$. Sparse Subspace Clustering (SSC) [15] assumes that the data points lie in a low-dimensional subspace and seeks to find a sparse affinity matrix with $\mathcal{R}(C) = \|C\|_1 = \sum_{i,j=1}^n |c_{ij}|$ under a constraint $\mathrm{diag}(C) = 0$. Low-Rank Representation (LRR) [16] is another approach that seeks a low-rank affinity matrix with $\mathcal{R}(C) = \|C\|_*$ (nuclear norm of $C$). Least-Squares Representation (LSR) [17, 18] involves solving a least-squares problem to find the representation matrix with $\mathcal{R}(C) = \frac{1}{2}\|C\|_F^2$. Their kernel variants, i.e., KSSC [19], KLRR [20], and KLSR [3] are used to extend their applicability to non-linearly separable data by applying a kernel function to data points with $\min_C \frac{1}{2}\|\phi(X) - \phi(X)C\|_F^2 + \lambda\mathcal{R}(C)$, showing promising performance in spectral clustering.

### 2.2 Missing Data Processing Techniques

**Data Imputation** is a popular technique [21, 22, 23] for dealing with incomplete data by filling in missing values. Statistical imputation methods, such as zero, mean imputation, and $k$-nearest neighbors ($k$NN) approach [26], have been widely used in practice. These methods replace the missing value with a zero, mean, or $k$-weighted value of non-missing elements in the corresponding feature. Additionally, matrix completion [27, 28, 29, 30] is a machine learning-based technique that fills missing values by solving a matrix factorization problem under assumptions on data structures such as low-rank or high-rank. However, correctly estimating missing values based on observed data is difficult, especially for a large missing ratio, and there is no guarantee on the quality of the affinity matrix on imputed data. This motivates us to design imputation-free approaches in Sections 3 and 4.

**Distance Calibration** is a specialized approach to obtaining a valid distance metric from an initial non-metric distance matrix, which can be applied to incomplete data. For any two incomplete data samples $x_i, x_j \in \mathbb{R}^d$, a new vector $x_i(I) \in \mathbb{R}^{|I|}$ is formed by selecting the observed values of $x_i$ on the set $I$, where $I$ is an index set of all features observed in both samples. The pairwise Euclidean distance between $x_i$ and $x_j$ can then be heuristically estimated by [39, 41]

$$d_{ij}^0 = \|x_i(I) - x_j(I)\|_2 \cdot \sqrt{\frac{d}{|I|}}. \tag{1}$$

However, the initial Euclidean distance matrix $D^0 = [d_{ij}^0] \in \mathbb{R}^{n \times n}$ estimated on incomplete data is usually not a distance metric due to missing values. Distance calibration methods [34, 35, 38, 39] can correct $D^0$ to a distance metric by making different assumptions and leveraging various properties. More details of these methods are discussed in Section 3.1.

## 3 Methodology-I. Kernel Correction

### 3.1 Revisiting Distance Calibration Methods

We begin with the definition of a distance metric [42], then delve into distance calibration methods, discussing the assumptions and properties underlying each method, as well as their limitations.

**Definition 1.** *A distance metric is defined as a $n \times n$ real symmetric matrix $D$ that satisfies*

$$d_{ij} = d_{ji} \geq 0, \ d_{ii} = 0, \ d_{ik} \leq d_{ij} + d_{jk}, \textit{ for all } 1 \leq i, j, k \leq n.$$

The properties of non-negativity, reflexivity, symmetry, and triangle inequality ensure that the distance metric produces meaningful and valuable distance measurements.

- **The Triangle Fixing (TRF) Algorithm** [35] to obtain a distance metric is based on the metric nearness model [34, 36], which finds the nearest approximation of a non-metric $D^0$ by solving:

$$\underset{D \in \mathbb{R}^{n \times n}}{\text{minimize}} \ \|D - D^0\|_F^2, \ \text{subject to} \ d_{ij} = d_{ji} \geq 0, \ d_{ii} = 0, \ d_{ik} \leq d_{ij} + d_{jk}, \ \forall \, 1 \leq i, j, k \leq n. \quad (2)$$

However, there are two significant limitations. First, the scalability of the algorithm is severely limited by the matrix size $n$. Since the number of $O(n^3)$ constraints grows rapidly as $n$ increases, the optimization time can become lengthy, taking several hours for a few thousand samples. Second, the performance of the algorithm depends on the extent of violation of the triangle inequalities in the initial distance matrix $D^0$. With a small missing ratio of features, $D^0$ already satisfies most triangle inequalities, i.e., $d_{ik}^0 \leq d_{ij}^0 + d_{jk}^0$, then the algorithm typically yields only marginal improvement.

- **The Double-Centering (DC) Algorithm** [38] starts with the Euclidean Distance Matrix (EDM) [43] and utilizes Property 1 [44], a well-known result in classical Multi-Dimensional Scaling (cMDS) [45], to build the connection between the similarity matrix and the EDM.

**Definition 2.** *A $n \times n$ real symmetric matrix $D$ is called an EDM if there exists $x_1, x_2, \ldots, x_n \in \mathbb{R}^d$ such that $d_{ij} = \|x_i - x_j\|^2$ for $i, j = 1, 2, \ldots, n$. (Note that it uses squared Euclidean distance.)*

**Property 1.** *A $n \times n$ real symmetric matrix $D$ is EDM if and only if $\text{diag}(D) = 0$, $S := -\frac{1}{2}JDJ \succeq 0$, $J := I - ee^\top/n$, where $I$ is the identity matrix and $e$ is the $n$-dimension vector of all ones.*

The DC algorithm first finds the nearest positive semi-definite $\hat{S}$ by solving

$$\underset{S \in \mathbb{R}^{n \times n}}{\text{minimize}} \ \|S - (-\frac{1}{2}J(D^0 \circ D^0)J)\|_F^2, \ \text{subject to} \ S \succeq 0, \quad (3)$$

where $D^0$ is the initial Euclidean distance matrix and $\circ$ denotes the Hadamard product. The algorithm then transforms $\hat{S}$ into an EDM $\hat{D}$ using $\hat{d}_{ij} = \hat{s}_{ii} + \hat{s}_{jj} - 2\hat{s}_{ij}$, and obtain the calibrated (non-squared) distance matrix by $\hat{D} \leftarrow [\hat{d}_{ij}^{1/2}]$, but unfortunately, the quality of $\hat{D}$ cannot be guaranteed, and important information may be lost during the transformation [38].

- **The Euclidean Embedding (EE) Algorithm** [39, 40, 41] leverages Euclidean embeddable property [39, 46] to obtain a calibrated embeddable distance matrix by solving

$$\underset{D \in \mathbb{R}^{n \times n}}{\text{minimize}} \ \|D - D^0\|_F^2, \ \text{subject to} \ \exp(-\gamma D) \succeq 0, \ d_{ii} = 0, \ d_{ij} = d_{ji} \geq 0, \ \forall \, 1 \leq i, j \leq n. \quad (4)$$

**Definition 3.** *A $n \times n$ real symmetric matrix $D$ is said Euclidean embeddable if there exists $x_1, \ldots, x_n$ in Euclidean space and a distance function $\rho$ such that $d_{ij} = \rho(x_i, x_j) \geq 0$ for $i, j = 1, \ldots, n$.*

**Property 2.** *[Theorem 2, [39]] If a $n \times n$ real symmetric matrix $D$ is Euclidean embeddable, then the kernel $K := \exp(-\gamma D)$ is positive semi-definite for any $\gamma > 0$.*

It is important to note that these methods can only provide a calibrated distance matrix with benefits for distance-based kernels and are not a universal solution for dealing with incomplete data in spectral clustering tasks, which motivates us to further design a kernel-specialized method in Section 3.2.

### 3.2 Kernel Correction Algorithm

To overcome the limitations of distance calibration, we propose a new kernel correction algorithm that directly focuses on the construction of a high-quality kernel with a theoretical guarantee. We consider an incomplete data matrix $X \in \mathbb{R}^{d \times n}$ with missing values and an initial kernel matrix $K^0 \in \mathbb{R}^{n \times n}$ estimated from $X$. Inspired by our previous work [31, 47, 48], our goal is to correct the initial kernel $K^0$ to an improved estimate $\hat{K}$ that satisfies the PSD property based on the Lemma 1 [49].

**Lemma 1.** *A valid kernel matrix is a $n \times n$ real symmetric matrix that satisfies the PSD property.*

Naturally, we recover the PSD property with the minimum cost by solving the following model:

$$\underset{K \in \mathbb{R}^{n \times n}}{\text{minimize}} \ \|K - K^0\|_F^2, \ \text{subject to} \ K \succeq 0, \ k_{ij} = k_{ji} \in [l, u], \ \forall \, 1 \leq i, j \leq n, \quad (5)$$

where $l, u$ denote the lower bound and upper bound, respectively.

It is worth noting that the solution $\hat{K}$ to Eq. (5) provides an improved estimate of the unknown ground-truth $K^*$ compared to $K^0$, as illustrated in Theorem 1. The proof comes from Kolmogorov's criterion [50, 31, 47, 48], which characterizes the best estimation in an inner product space, provided in Appendix A.

**Theorem 1.** $\|K^* - \hat{K}\|_F \leq \|K^* - K^0\|_F$. *The equality holds if and only if $K^0 \succeq 0$, i.e., $K^0 = \hat{K}$.*

Regarding the kernel type, Gaussian kernel is a widely-used non-linear kernel that has elements in the range $[0, 1]$. In this case, the feasible region in Eq. (5) is defined as $\mathcal{F} = \{K \in \mathbb{R}^{n \times n} \mid K \succeq 0, k_{ii} = 1, k_{ij} = k_{ji} \in [0, 1], \forall 1 \leq i, j \leq n\}$, which is a closed convex set. The solution to Eq. (5) is the projection of $K^0$ onto $\mathcal{F}$, denoted by $\hat{K}$. However, finding the direct projection is complex, and no closed form of $\hat{K}$ exists. Thus, we break down $\mathcal{F}$ into two simpler, closed convex subsets $\mathcal{F}_1$ and $\mathcal{F}_2$: $\mathcal{F}_1 = \{K \in \mathbb{R}^{n \times n} \mid K \succeq 0\}$, $\mathcal{F}_2 = \{K \in \mathbb{R}^{n \times n} \mid k_{ii} = 1, k_{ij} = k_{ji} \in [0, 1], \forall 1 \leq i, j \leq n\}$, with $\mathcal{F} = \mathcal{F}_1 \cap \mathcal{F}_2$. Then $\hat{K}$ can be efficiently solved by iteratively projecting $K^0$ onto $\mathcal{F}_1$ and $\mathcal{F}_2$ [31, 40, 47]. Denote $\mathcal{P}_1, \mathcal{P}_2$ as the projection onto $\mathcal{F}_1, \mathcal{F}_2$, respectively, in the form of

$$\begin{cases} \mathcal{P}_1(K) = U\hat{\Sigma}U^\top \text{ where } K = U\Sigma U^\top, \ \hat{\Sigma}_{ij} = \max\{\Sigma_{ij}, 0\}, \\ \mathcal{P}_2(K) = [\mathcal{P}_2(k_{ij})] \text{ where } \mathcal{P}_2(k_{ij}) = \text{median}\{0, k_{ij}, 1\}, \ \mathcal{P}_2(k_{ii}) = 1, \end{cases} \quad (6)$$

where $U\Sigma U^\top$ gives the spectral decomposition (SD) of $K$.

We use Dykstra's projection algorithm [51] to find the optimal projection, summarized in Algorithm 1:

---
**Algorithm 1 K**ernel **C**orrection (**KC**)
---
**Input:** $K^0 \in \mathbb{R}^{n \times n}$: an initial (non-PSD) kernel matrix; $\mathcal{P}_1, \mathcal{P}_2$: the projection onto $\mathcal{F}_1$ and $\mathcal{F}_2$;     $maxiter$: maximum iterations (default 100); $tol$: tolerence (default $10^{-5}$).
**Output:** $\hat{K} \in \mathbb{R}^{n \times n}$: the optimal corrected kernel matrix.
1: ▷ *Dykstra's Projection: $Y, P, Q$ are auxiliary variables.*
2: Initialize $X_0 = K^0$ and $P_0 = Q_0 = \mathbf{0}_{n \times n}$.
3: **for** $t = 0, 1, \ldots, maxiter$ **do**
4:     $Y_t = \mathcal{P}_2(X_t + P_t)$,
5:     $P_{t+1} = X_t + P_t - Y_t$,
6:     $X_{t+1} = \mathcal{P}_1(Y_t + Q_t)$,
7:     $Q_{t+1} = Y_t + Q_t - X_{t+1}$.
8:     **if** $\|X_{t+1} - X_t\|_F < tol$ **then**
9:        **break**
10:     **end if**
11: **end for**
12: Set $\hat{K} = X_t$.
---

The convergence guarantee of Algorithm 1 relies on the Boyle-Dykstra's result [52]:

**Lemma 2.** *Given a real symmetric matrix $K^0 \in \mathbb{R}^{n \times n}$, the sequence $\{X_t\}$ generated in Algorithm 1 converges to $\hat{K} = \min_{K \in \mathcal{F} = \mathcal{F}_1 \cap \mathcal{F}_2} \|K - K^0\|_F^2$ as $t \to \infty$.*

### 3.3 Limitation and Complexity Analysis

The potential limitation primarily stems from the time complexity of Algorithm 1. The pre-iteration time complexity of the KC algorithm is currently at $O(n^3)$, which mainly arises from the spectral decomposition (SD) in the projection operation $\mathcal{P}_1$ and poses challenges when dealing with large-scale datasets. To address this issue, a possible solution is to replace the spectral decomposition with a randomized singular value decomposition (rSVD) [53]. The rSVD approach seeks top-$k$ singular values and effectively reduces the time complexity to $O(n^2 \cdot \log(k) + 2n \cdot k^2)$. However, the trade-off between efficiency and efficacy necessitates further investigation. Besides, we can transform Dykstra's projection into a parallel version with cyclic projection [54] to achieve better scalability. The storage complexity is $O(n^2)$ to store the dense kernel matrix in memory.

### 3.4 Discussion on Kernel Correction

The proposed kernel correction approach provides an imputation-free method with several benefits to spectral clustering, compared to data imputation and distance calibration methods.

**Differences with Previous Kernel Methods** Some literature in kernel learning [55, 56] has studied missing data in a supervised manner. However, our work focuses on unsupervised learning and addresses *complete but inaccurate* (noisy) kernels due to the presence of incomplete observations, which is fundamentally different from previous work [57] primarily dealing with incomplete kernels.

**Advantages over Data Imputation** Our approach eliminates the need for domain knowledge in handling incomplete data by bypassing the imputation step. This enables us to generate a kernel matrix that is theoretically guaranteed, offering potential advantages for various kernel-based applications. Notably, our approach demonstrates significant improvements over imputation methods for spectral clustering when dealing with a high proportion of missing data. Additionally, the high-quality kernel produced by our approach can serve as a valuable reference for improving the accuracy of missing value estimation in kernel-based imputation methods [58, 59], which is worth further investigation.

**Advantages over Distance Calibration** Firstly, our approach can be applied to a wide range of kernels and yields an improved kernel matrix. In contrast, distance calibration methods only benefit distance-based kernels and lack quality guarantees. Moreover, our algorithm, which corrects the Gaussian kernel, can generate a high-quality distance matrix via $d_{ij} = \sqrt{-\sigma^2 \log(k_{ij})}$. By incorporating the second-order term of $d_{ij}^0$ in $k_{ij}^0 = \exp(-(d_{ij}^0)^2/\sigma^2)$, our algorithm becomes more sensitive to changes in distance values, while the Euclidean embedding algorithm relies on $K^0 = \exp(-\gamma D^0)$ using the first-order term of $d_{ij}^0$. Empirical evidence suggests that the corrected distance obtained from our corrected Gaussian kernel is more accurate than the calibrated distance derived from the Euclidean embedding method, benefiting to the distance-based spectral clustering.

**Benefits to Spectral Clustering** Our approach is tailored to improve the kernel quality, making it highly advantageous for spectral clustering. This improvement benefits both standard spectral clustering using the Gaussian kernel and affinity learning based on the self-expressive framework. In the case of $X$-based self-expressive affinity, we can apply the corrected linear kernel $K := X^\top X$ to the optimization, i.e., $\min_C \frac{1}{2}\|X - XC\|_F^2 + \lambda \mathcal{R}(C) = \min_C \frac{1}{2}\text{Tr}(K - 2KC + C^\top KC) + \lambda \mathcal{R}(C)$. Similarly, for $K$-based self-expressive affinity, which involves using the Gaussian kernel or other kernels, our approach can be applied to $K := \phi(X)^\top \phi(X)$ on $\min_C \frac{1}{2}\|\phi(X) - \phi(X)C\|_F^2 + \lambda \mathcal{R}(C)$.

## 4 Methodology-II. Affinity Learning

Taking advantage of the corrected kernel matrix, we have further designed a series of kernel-based affinity learning algorithms to acquire high-quality affinity matrices with an adaptive extension.

### 4.1 Kernel Self-expressive Learning with $\ell_p$-norm

We utilize the kernel self-expressive framework, i.e., $\min_C \frac{1}{2}\|\phi(X) - \phi(X)C\|_F^2 + \lambda \mathcal{R}(C)$, and propose new affinity learning methods by enhancing the sparsity of the affinity matrix $C$ using the $\ell_p$ norm [60]. Specifically, our approach diverges from previous work [19, 20, 3] by incorporating two forms of $\ell_p$-norm, i.e., proximal $p$-norm and Schatten $p$-norm ($0 < p < 1$), as regularization terms.

**Definition 4 (Kernel Self-expressive Learning with Proximal $p$-norm).**

$$\underset{C \in \mathbb{R}^{n \times n}}{\text{minimize}} \quad \frac{1}{2}\|\phi(X) - \phi(X)C\|_F^2 + \frac{\lambda}{2}\|C\|_p^p, \ \text{ subject to } \ 0 \leq c_{ij} \leq 1, \ \forall \ 1 \leq i, j \leq n, \quad (7)$$

*where the proximal $p$-norm is expressed as $\|C\|_p^p := \sum_{i,j=1}^n |c_{ij}|^p$.*

We construct an augmented Lagrangian function $\mathcal{L}_p(C, Z, U)$ in Eq. (8), and solve it by using the Alternating Direction Method of Multipliers (ADMM) approach [61] summarized in Algorithm 2.

$$\mathcal{L}_p = \|\phi(X) - \phi(X)C\|_F^2 + \lambda\|Z\|_p^p + \gamma \sum_{i,j} \max(z_{ij} - 1, 0)^2 + \text{Tr}(U^\top(Z - C)) + \frac{1}{2\rho}\|Z - C\|_F^2. \quad (8)$$

However, the KSL-Pp algorithm involves numerous hyper-parameters and entails a non-convex optimization process during the $Z$-update step (Line 5 in Algorithm 2), making it difficult to effectively utilize. Computation speedup needs further investigations. Thus, we focus on the Schatten $p$-norm.

---
**Algorithm 2** **K**ernel **S**elf-expressive **L**earning with **P**roximal $p$-norm (**KSL-Pp**)
---
**Input:** $K$: a kernel; $\lambda, p, \rho, \gamma, \alpha$: hyperparameters; $maxiter$: maximum iterations; $tol$: tolerance.
**Output:** $\hat{C} \in \mathbb{R}^{n \times n}$: the optimal affinity matrix of Eq. (7).
1: Initialize $C_0, Z_0, U_0$.                 ▷ *Refer to Appendix B.1 for formula derivations.*
2: **for** $t = 0, 1, \ldots, maxiter$ **do**
3:    $C$-update: $C_{t+1} \leftarrow (2K + \frac{1}{\rho}I)^{-1}(2K + U_t + \frac{1}{\rho}Z_t)$;
4:    $Z$-update: $Z_{t+1} \leftarrow \frac{\partial \mathcal{L}_p}{\partial z_{ij}} = \lambda p z_{ij}^{p-1} + \gamma \mathbb{I}_{(z_{ij} > 1)} + \frac{1}{\rho}(z_{ij} - c_{ij}) + u_{ij} = 0$;
5:    $U$-update: $U_{t+1} \leftarrow U_t + \rho(Z_{t+1} - C_{t+1})$.
6:    **if** (primal) $\|C_{t+1} - z_{t+1}\|_F \leq tol$ and (dual) $\|Z_{t+1} - Z_t\|_F \leq tol$ **then**;   **break**;   **end if**.
7: **end for**
8: Set $\hat{C} = (|C_t| + |C_t^\top|)/2$.
---

**Definition 5** (**Kernel Self-expressive Learning with Schatten $p$-norm**).

$$\underset{C \in \mathbb{R}^{n \times n}}{\text{minimize}} \ \frac{1}{2}\|\phi(X) - \phi(X)C\|_F^2 + \frac{\lambda}{2}\|C\|_{S_p}, \ \text{subject to} \ 0 \leq c_{ij} \leq 1, \ \forall \ 1 \leq i, j \leq n, \quad (9)$$

*where Schatten p-norm* $\|C\|_{S_p} := (\sum_{i=1}^n \sigma_i^p(C))^{1/p}$ *and* $\sigma_i(C)$ *denotes the $i$-th singular value of $C$.*

Drawing from prior research [62], for $\frac{1}{2} < p < 1$ and $k \geq \text{rank}(C)$, the following always holds true:

$$\|C\|_{S_p} = \underset{U \in \mathbb{R}^{n \times k}, V \in \mathbb{R}^{n \times k}, C = UV^\top}{\text{minimize}} \frac{\|U\|_F^2 + \|V\|_F^2}{2}.$$

Thus, for $\frac{1}{2} < p < 1$, we define $\mathcal{L}_{S_p}(U, V) = \|\phi(X) - \phi(X)UV^\top\|_F^2 + \frac{\lambda}{2}\|U\|_F^2 + \frac{\lambda}{2}\|V\|_F^2$ as a relaxation of the optimization problem in Eq. (9) without loss of generality. By employing the ADMM approach [61] and the gradient descent method [63] in the $U$-update step (i.e., Line 4 in Algorithm 3), we design the KSL-Sp algorithm as follows with details in the Appendix B.2.

---
**Algorithm 3** **K**ernel **S**elf-expressive **L**earning with **S**chatten $p$-norm (**KSL-Sp**)
---
**Input:** $K$: a kernel matrix; $\lambda$: a hyperparameter; $maxiter$: maximum iterations; $tol$: tolerance.
**Output:** $\hat{C} \in \mathbb{R}^{n \times n}$: the optimal affinity matrix of Eq. (9).
1: Initialize $U_0, V_0$.                 ▷ *Refer to Appendix B.2 for formula derivations.*
2: **for** $t = 0, 1, \ldots, maxiter$ **do**
3:    $U$-update: $U_{t+1} \leftarrow \partial_U \mathcal{L}_{S_p} = 2K(UV_t^\top - I)V_t + \lambda U = 0$;
4:    $V$-update: $V_{t+1} \leftarrow 2KU_{t+1}(2U_{t+1}^\top KU_{t+1} + \lambda I)^{-1}$ by $\partial_V \mathcal{L}_{S_p} = 0$.
5:    **if** $\|U_{t+1} - U_t\|_F < tol$ **then**;   **break**;   **end if**.
6: **end for**
7: Set $C_t = U_t V_t^\top$ and $\hat{C} = (|C_t| + |C_t^\top|)/2$.
---

## 4.2 Adaptive Kernel Self-expressive Learning

To achieve adaptive affinity learning on incomplete data, we formulate a new joint optimization problem by incorporating KLSR [3] and kernel correction to learn kernel and affinity matrices iteratively with the PSD constraint of $K$:

$$\underset{K \succeq 0, C \in \mathbb{R}^{n \times n}}{\text{minimize}} \ \|K - K^0\|_F^2 + \text{Tr}(K - 2KC + C^\top KC) + \lambda\|C\|_F^2. \quad (10)$$

Firstly, we introduce an auxiliary variable $A \in \mathbb{R}^{n \times n}$ into

$$\underset{K \succeq 0, A, C \in \mathbb{R}^{n \times n}}{\text{minimize}} \ \|K - K^0\|_F^2 + \text{Tr}(A - 2AC + C^\top AC) + \lambda\|C\|_F^2, \ \text{subject to} \ K = A, \quad (11)$$

and then we derive the augmented Lagrange function $\mathcal{L}(K, A, C, U)$ in preparation for ADMM:

$$\mathcal{L} = \|K - K^0\|_F^2 + \text{Tr}(A - 2AC + C^\top AC) + \lambda\|C\|_F^2 + \text{Tr}(U^\top(K - A)) + \frac{\rho}{2}\|K - A\|_F^2, \quad (12)$$

where $U$ is a Lagrange multiplier and $\rho$ is the updating step size, finally arriving at the Algorithm 4.

Our adaptive learning framework could be combined with other kernel self-expressive learning algorithms; however, due to space constraints, further research is required to explore these possibilities:

$$\underset{K \succeq 0, C \in \mathbb{R}^{n \times n}}{\text{minimize}} \quad \|K - K^0\|_F^2 + \lambda_1 \|\phi(X) - \phi(X)C\|_F^2 + \lambda_2 \mathcal{R}(C), \text{ where } K := \phi(X)^\top \phi(X). \quad (13)$$

---

**Algorithm 4** **A**daptive **K**ernel **L**east-**S**quares **R**epresentation (**AKLSR**)

---

**Input:** $K^0$: an initial kernel; $\lambda, \rho$: hyperparameters; $maxiter$: maximum iterations; $tol$: tolerance.
**Output:** $\hat{C} \in \mathbb{R}^{n \times n}$: the optimal affinity matrix of Eq. (10).
1: Initialize $K_0, A_0, C_0, U_0$.                 ▷ *Refer to Appendix B.3 for formula derivations.*
2: **for** $t = 0, 1, \ldots, maxiter$ **do**
3:      $K$-update: $K \leftarrow \frac{1}{\rho+2}(2K^0 - U_t + \rho A_t)$ by $\frac{\partial \mathcal{L}}{\partial K} = 0$;
4:      $K$-PSD: $K_{t+1} \leftarrow \mathcal{P}_1(K)$ by projecting $K$ onto the PSD set via Eq. (6);
5:      $A$-update: $A_{t+1} \leftarrow \frac{1}{\rho}(\rho K_{t+1} + U_t + 2C_t^\top - C_t C_t^\top - I)$ by $\frac{\partial \mathcal{L}}{\partial A} = 0$;
6:      $C$-update: $C_{t+1} \leftarrow 2(2\lambda I + A_{t+1} + A_{t+1}^\top)^{-1} A_{t+1}^\top$ by $\frac{\partial \mathcal{L}}{\partial C} = 0$;
7:      $U$-update: $U_{t+1} \leftarrow U_t + \rho(K_{t+1} - A_{t+1})$.
8:      **if** $\|C_{t+1} - C_t\|_F < tol$ **then**;    **break**;    **end if**.
9: **end for**
10: Set $\hat{C} = (|C_t| + |C_t^\top|)/2$.

---

## 5 Experiments

We evaluate the performance on four benchmark datasets, including two face image datasets **Yale64** [64] and **Umist** [65], a handwritten digit image dataset **USPS** [66], and a speech dataset **Isolet** [67]. We use a subset of USPS with 1,000 randomly selected samples. All experiments [2] are conducted five times in MATLAB on a ThinkStation with a 2.1 GHz Intel i7-12700 Core and 32GB RAM.

Various methods dealing with incomplete data are considered for comparison: 1) statistical imputation: **ZERO**, **MEAN**, $k$-nearest neighbors (**kNN**) [26], Expectation Maximization (**EM**) [68]; 2) matrix completion: Singular Value Thresholding (**SVT**) [27], Grassmanian Rank-one Update Subspace Estimation (**GR**) [28], Kernelized Factorization Matrix Completion (**KFMC**) [29]; 3) distance calibration: Double-Centering (**DC**) [38], Triangle Fixing (**TRF**) [35], Euclidean Embedding (**EE**) [39]. Our KC method uses the spectral decomposition (SD). Details are provided in Appendix C.

### 5.1 Validation of Kernel Correction on Gaussian Kernel and Euclidean Distance

When handling incomplete data that is missing completely at random, the quality of the estimated Gaussian kernel matrix and Euclidean distance matrix is measured using the relative-mean-square error (**RMSE** $= \frac{\|\hat{A} - A^*\|_F^2}{\|A^0 - A^*\|_F^2}$) and the relative error (**RE** $= \frac{\|\hat{A} - A^*\|_F}{\|A^*\|_F}$), where $A^0$ is the naive kernel/distance matrix obtained by Eq. (1), $\hat{A}$ is the estimated one using different methods, and $A^*$ as the ground-truth. All Gaussian kernels are calculated by $k_{ij} = \exp(-d_{ij}^2/\sigma^2)$, where $\sigma = \text{median}\{d_{ij}\}$. Our KC algorithm yields Gaussian kernel and Euclidean distance matrices with the lowest RMSEs and REs, validating the theoretical guarantee of kernel matrices in Theorem 1, as demonstrated in Table 1. Furthermore, the accuracy of the top-10 nearest neighbors is evaluated using Recall values [69]. The KC algorithm achieves the highest Recalls with improved local relationships, which in turn benefits spectral clustering, particularly for $k$NN graphs. More numerical results are in Appendix D.

Table 1: Quality comparisons of Gaussian kernel and Euclidean distance under a missing ratio 80%.

| Dataset | Metric | Naive | ZERO | MEAN | $k$NN | EM | SVT | GR | KFMC | DC | TRF | EE | KC |
|---|---|---|---|---|---|---|---|---|---|---|---|---|---|
| | RMSE-K↓ | 1.000 | 2.849 | 1.535 | 2.451 | 1.527 | 2.849 | 1.797 | 3.818 | 1.748 | 0.923 | 0.438 | **0.382** |
| | RMSE-D↓ | 1.000 | 20.80 | 27.64 | 21.20 | 27.62 | 20.80 | 8.397 | 14.05 | 110.4 | 0.916 | 0.440 | **0.431** |
| Umist | RE-K↓ | 0.189 | 0.319 | 0.234 | 0.295 | 0.233 | 0.319 | 0.253 | 0.369 | 0.250 | 0.181 | 0.125 | **0.117** |
| | RE-D↓ | 0.107 | 0.487 | 0.561 | 0.492 | 0.561 | 0.487 | 0.309 | 0.400 | 1.122 | 0.102 | 0.071 | **0.070** |
| | Recall↑ | 0.726 | 0.092 | 0.171 | 0.119 | 0.172 | 0.092 | 0.596 | 0.248 | 0.226 | 0.740 | 0.771 | **0.785** |

---

[2]Codes are available at https://github.com/SciYu/Spectral-Clustering-on-Incomplete-Data.

## 5.2 Comparative Studies on Standard Spectral Clustering

We adopt the Gaussian kernel for standard spectral clustering and obtain its $k$NN graph ($k = 10$), as the input for clustering via the normalized cut algorithm [1]. We evaluate the clustering performance using Accuracy (**ACC**), Normalized Mutual Information (**NMI**), Purity (**PUR**), and Adjusted Rand Index (**ARI**). Table 2 shows that our KC algorithm, which corrects the Gaussian kernel, achieves the best clustering performance under a large missing ratio $80\%$ (i.e., missing completely at random).

Table 2: Performance of standard spectral clustering on incomplete data under a missing ratio $80\%$.

| Dataset | Metric | Naive | ZERO | MEAN | $k$NN | EM | SVT | GR | KFMC | DC | TRF | EE | KC |
|---|---|---|---|---|---|---|---|---|---|---|---|---|---|
| Yale64 | ACC | 0.561 | 0.218 | 0.365 | 0.227 | 0.374 | 0.224 | 0.218 | 0.259 | 0.513 | 0.562 | 0.573 | **0.578** |
| | NMI | 0.588 | 0.246 | 0.429 | 0.257 | 0.428 | 0.269 | 0.264 | 0.297 | 0.551 | 0.587 | 0.593 | **0.596** |
| | PUR | 0.572 | 0.233 | 0.390 | 0.241 | 0.396 | 0.241 | 0.232 | 0.273 | 0.525 | 0.570 | 0.581 | **0.584** |
| | ARI | 0.353 | 0.010 | 0.137 | 0.015 | 0.145 | 0.017 | 0.012 | 0.035 | 0.293 | 0.350 | 0.357 | **0.366** |
| Umist | ACC | 0.462 | 0.220 | 0.351 | 0.230 | 0.349 | 0.218 | 0.410 | 0.314 | 0.350 | 0.462 | 0.461 | **0.463** |
| | NMI | 0.669 | 0.282 | 0.478 | 0.314 | 0.479 | 0.286 | 0.597 | 0.423 | 0.488 | 0.667 | 0.669 | **0.673** |
| | PUR | 0.549 | 0.245 | 0.415 | 0.261 | 0.419 | 0.247 | 0.502 | 0.366 | 0.408 | 0.546 | 0.551 | **0.553** |
| | ARI | 0.373 | 0.070 | 0.206 | 0.082 | 0.207 | 0.067 | 0.304 | 0.140 | 0.216 | 0.370 | 0.371 | **0.377** |
| USPS | ACC | 0.343 | 0.350 | 0.362 | 0.360 | 0.222 | 0.353 | 0.375 | 0.351 | 0.418 | 0.464 | 0.511 | **0.523** |
| | NMI | 0.222 | 0.228 | 0.278 | 0.265 | 0.104 | 0.236 | 0.319 | 0.358 | 0.312 | 0.393 | 0.457 | **0.472** |
| | PUR | 0.395 | 0.400 | 0.434 | 0.427 | 0.260 | 0.402 | 0.440 | 0.460 | 0.473 | 0.535 | 0.594 | **0.609** |
| | ARI | 0.168 | 0.164 | 0.180 | 0.175 | 0.051 | 0.167 | 0.206 | 0.178 | 0.231 | 0.304 | 0.344 | **0.360** |
| Isolet | ACC | 0.495 | 0.260 | 0.245 | 0.297 | 0.172 | 0.263 | 0.357 | 0.324 | 0.243 | 0.515 | 0.560 | **0.561** |
| | NMI | 0.613 | 0.339 | 0.330 | 0.383 | 0.228 | 0.342 | 0.492 | 0.465 | 0.323 | 0.643 | 0.660 | **0.672** |
| | PUR | 0.520 | 0.278 | 0.261 | 0.318 | 0.183 | 0.277 | 0.377 | 0.354 | 0.260 | 0.542 | 0.584 | **0.593** |
| | ARI | 0.364 | 0.126 | 0.108 | 0.159 | 0.055 | 0.131 | 0.210 | 0.177 | 0.105 | 0.387 | 0.429 | **0.432** |

We also examine robustness across varying missing ratios. Fig. 1 shows the KC method's advantage over imputation, especially at large missingness. Numerical results are in Appendix E.

• The KC method consistently surpasses others in kernel estimation, evidenced by reduced relative errors (RE-K) and enhanced neighborhood ties in the $k$NN graph, seen through higher Recall values.

• At small missing ratios, the KC method offers incremental improvements over imputation. However, at large missing ratios, imputation methods falter due to scant observed data, leading to increased errors and compromised clustering. Conversely, the KC method maintains stable clustering outcomes.

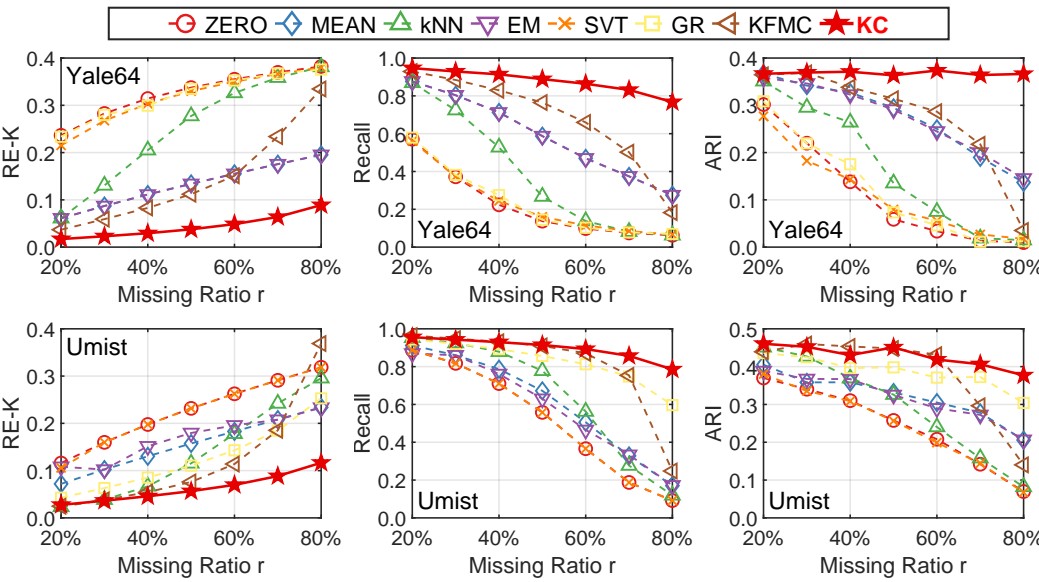

Figure 1: Robustness analysis of kernel estimation, neighborhood relationship, and standard spectral clustering on the Yale64 and Umist datasets under a wide range of missing ratios, i.e., $r \in [20\%, 80\%]$.

## 5.3 Comparative Studies on Self-expressive Affinity Learning

We delve deeper into the advantages of the corrected Gaussian kernel in kernel self-expressive affinity learning, encompassing methods like KSSC [19], KLSR [3], KSL-Sp, and AKLSR, with KSL-Pp excluded due to its impracticality. As Table 3 illustrates, our KC method excels over baselines in the NMI. Notably, both KSL-Sp and AKLSR algorithms employ corrected kernels to yield dependable affinity matrices, thereby elevating performance. Furthermore, our imputation-free framework could augment spectral clustering using other kernels, such as the Exponential kernel. A comprehensive comparison and in-depth analysis are available in Appendix F. Collectively, our study presents a holistic approach to clustering incomplete data, paving the way for subsequent research.

Table 3: NMI performance of self-expressive affinity on incomplete data under a missing ratio 80%.

| Method | Dataset | Naive | ZERO | MEAN | $k$NN | EM | SVT | GR | KFMC | DC | TRF | EE | KC |
|--------|---------|-------|------|------|-------|-----|-----|-----|------|-----|-----|-----|-----|
| KSSC | Yale64 | 0.219 | 0.215 | 0.167 | 0.173 | 0.177 | 0.218 | 0.208 | 0.259 | 0.588 | 0.210 | 0.209 | **0.616** |
| | Umist | 0.101 | 0.151 | 0.198 | 0.154 | 0.178 | 0.141 | 0.639 | 0.254 | 0.123 | 0.121 | 0.139 | **0.714** |
| | USPS | 0.017 | 0.196 | 0.396 | 0.271 | 0.222 | 0.200 | 0.360 | 0.359 | 0.077 | 0.018 | 0.025 | **0.410** |
| | Isolet | 0.065 | 0.024 | 0.153 | 0.384 | 0.382 | 0.022 | 0.454 | 0.292 | 0.171 | 0.064 | 0.062 | **0.582** |
| KLSR | Yale64 | 0.606 | 0.311 | 0.604 | 0.320 | 0.609 | 0.321 | 0.327 | 0.318 | 0.597 | 0.603 | 0.604 | **0.616** |
| | Umist | 0.676 | 0.579 | 0.640 | 0.538 | 0.638 | 0.576 | 0.630 | 0.548 | 0.647 | 0.671 | 0.687 | **0.696** |
| | USPS | 0.019 | 0.225 | 0.417 | 0.317 | 0.235 | 0.231 | 0.356 | 0.360 | 0.418 | 0.019 | 0.420 | **0.485** |
| | Isolet | 0.215 | 0.398 | 0.548 | 0.463 | 0.400 | 0.400 | 0.551 | 0.512 | 0.522 | 0.306 | 0.628 | **0.659** |
| KSL-Sp | Yale64 | 0.370 | 0.315 | 0.581 | 0.303 | 0.579 | 0.305 | 0.304 | 0.295 | 0.555 | 0.364 | 0.599 | **0.619** |
| | Umist | 0.144 | 0.453 | 0.595 | 0.485 | 0.592 | 0.454 | 0.672 | 0.443 | 0.616 | 0.152 | 0.680 | **0.690** |
| | USPS | 0.073 | 0.288 | 0.360 | 0.342 | 0.196 | 0.285 | 0.331 | 0.439 | 0.238 | 0.055 | 0.517 | **0.524** |
| | Isolet | 0.266 | 0.410 | 0.556 | 0.487 | 0.364 | 0.410 | 0.485 | 0.557 | 0.383 | 0.246 | 0.624 | **0.634** |
| AKLSR | Yale64 | 0.452 | 0.327 | 0.606 | 0.338 | 0.605 | 0.308 | 0.338 | 0.312 | 0.570 | 0.464 | 0.575 | **0.614** |
| | Umist | 0.379 | 0.490 | 0.624 | 0.498 | 0.620 | 0.484 | 0.691 | 0.456 | 0.611 | 0.350 | 0.626 | **0.691** |
| | USPS | 0.021 | 0.114 | 0.336 | 0.235 | 0.236 | 0.115 | 0.098 | 0.017 | 0.327 | 0.031 | 0.259 | **0.365** |
| | Isolet | 0.100 | 0.246 | 0.448 | 0.381 | 0.370 | 0.246 | 0.069 | 0.072 | 0.453 | 0.119 | 0.411 | **0.496** |

## 5.4 Spectral Clustering on Block-missing Data: A Case Study

Beyond the experiments addressing data missing completely at random, we generate a block of appropriate sizes located randomly in images and values in the block are missing where the missingness is systematically related to locations. As illustrated in Table 4, under this block-missing paradigm [70], our KC method persistently outperforms existing data imputation and distance calibration techniques.

Table 4: NMI performance of self-expressive affinity on block-missing data under a missing ratio 80%. **Bold** font indicates the best, and underline indicates the second-best.

| Dataset | Method | Naive | ZERO | MEAN | $k$NN | EM | SVT | GR | KFMC | DC | TRF | EE | KC |
|---------|--------|-------|------|------|-------|-----|-----|-----|------|-----|-----|-----|-----|
| Yale64 | KSSC | 0.217 | 0.340 | 0.526 | 0.379 | 0.530 | 0.376 | 0.342 | 0.496 | 0.482 | 0.207 | 0.212 | **0.562** |
| | KLSR | 0.592 | 0.336 | 0.544 | 0.365 | 0.534 | 0.365 | 0.338 | 0.533 | 0.484 | **0.596** | 0.585 | 0.592 |
| | KSL-Sp | 0.488 | 0.335 | 0.540 | 0.367 | 0.544 | 0.366 | 0.343 | 0.543 | 0.481 | 0.484 | 0.584 | **0.591** |
| | AKLSR | 0.548 | 0.342 | 0.544 | 0.367 | 0.539 | 0.378 | 0.342 | 0.540 | 0.467 | 0.546 | 0.578 | **0.605** |

## 6 Conclusion and Future Work

In this paper, we propose an imputation-free framework for spectral clustering on incomplete data. Our framework directly learns high-quality affinity matrices via kernel correction and affinity learning, improving clustering performance. We introduce a novel kernel correction method with a theoretical guarantee, new affinity learning methods, and an adaptive extension that simultaneously learns high-quality kernel and affinity matrices. Extensive experiments demonstrate the effectiveness of our proposed methods on various benchmark datasets, showing superior performance compared to existing data imputation and distance calibration approaches. Our work provides a systematic solution to clustering incomplete data, with future work focused on integrating with deep clustering methods for various types of incomplete data and enhancing the speed and scalability of the algorithms.

## Acknowledgments

We appreciate the anonymous reviewers for their insightful feedback that greatly enhanced this paper. The work of Fangchen Yu and Wenye Li was supported in part by Guangdong Basic and Applied Basic Research Foundation (2021A1515011825), Guangdong Introducing Innovative and Entrepreneurial Teams Fund (2017ZT07X152), Shenzhen Science and Technology Program (CUHKSZWDZC0004), and Shenzhen Research Institute of Big Data Scholarship Program. The work of Jicong Fan was partially supported by the Youth program 62106211 of the National Natural Science Foundation of China. The work of Yicheng Zeng was supported by the Shenzhen Science and Technology Program (Grant No. RCBS20221008093336086), the Youth Program (Grant No. 12301383) of National Natural Science Foundation of China, and the Internal Project Fund from Shenzhen Research Institute of Big Data (Grant No. J00220230012). The work of Jianfeng Mao was supported in part by the National Natural Science Foundation of China (NSFC) under grant U1733102, in part by the National Natural Science Foundation of China (NSFC) under grant 72394362, in part by the Guangdong Provincial Key Laboratory of Big Data Computing, The Chinese University of Hong Kong, Shenzhen (CUHK-Shenzhen) under grant B10120210117, and in part by CUHK-Shenzhen under grant PF.01.000404.

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

# Supplementary Material

## Boosting Spectral Clustering on Incomplete Data via Kernel Correction and Affinity Learning

This document serves as supplementary material for the paper entitled "Boosting Spectral Clustering on Incomplete Data via Kernel Correction and Affinity Learning". It contains a theoretical analysis with proofs and formula derivation of theorems and algorithms presented in the main paper, a detailed description of experimental settings, and additional numerical results. Specifically,

**Sec. A** and **B** provide the proof of Theorem 1 and the formula derivations of Kernel Self-expressive Learning (KSL) algorithms with an adaptive extension, respectively;

**Sec. C** illustrates experimental settings and implementation details;

**Sec. D** validates the benefits of the kernel correction algorithm on kernel and distance estimation;

**Sec. E** and **F** present comprehensive results and in-depth analysis on clustering performance.

## A Theoretical Property of Kernel Correction

**Theorem 1.** $\|K^* - \hat{K}\|_F \leq \|K^* - K^0\|_F$. *The equality holds if and only if $K^0 \succeq 0$, i.e., $K^0 = \hat{K}$.*

*Proof.* Denote the feasible region as $\mathcal{F}$, which is a closed convex set. Let $K^0$ be the initial kernel matrix. Let $\hat{K}$ be the projection of $K^0$ onto $\mathcal{F}$ ($\hat{K} = \min_{K \in \mathcal{F}} \|K - K^0\|_F^2$). Let $K^*$ be the unknown ground-truth with positive semi-definiteness (PSD) in the feasible region $\mathcal{F}$. By definition, we have:

$$\|\hat{K} - K^0\|_F^2 \leq \|K^* - K^0\|_F^2.$$

Adding and subtracting $\hat{K}$ in the right-hand side, we get:

$$\|K^* - K^0\|_F^2 = \|K^* - \hat{K} + \hat{K} - K^0\|_F^2.$$

By the Pythagorean theorem [71], we have:

$$\|K^* - \hat{K} + \hat{K} - K^0\|_F^2 = \|K^* - \hat{K}\|_F^2 + \|\hat{K} - K^0\|_F^2 - 2\langle K^* - \hat{K}, K^0 - \hat{K}\rangle_F,$$

where $\langle \cdot, \cdot \rangle_F$ denotes the Frobenius inner product. Since $\hat{K}$ is the projection of $K^0$ onto the feasible region $\mathcal{F}$, we have $\langle K - \hat{K}, K^0 - \hat{K}\rangle_F \leq 0$ for all PSD matrices $K$ in $\mathcal{F}$ [72]. Therefore, we have:

$$\|K^* - K^0\|_F^2 = \|K^* - \hat{K} + \hat{K} - K^0\|_F^2 \geq \|K^* - \hat{K}\|_F^2 + \|\hat{K} - K^0\|_F^2.$$

If $K^0$ is already a PSD matrix in $\mathcal{F}$, we have:

$$\hat{K} = K^0 \ \text{ and } \ \|K^* - K^0\|_F^2 = \|K^* - \hat{K}\|_F^2.$$

Otherwise, we obtain:

$$\|\hat{K} - K^0\|_F^2 \geq 0,$$

which implies:

$$\|K^* - K^0\|_F^2 \geq \|K^* - \hat{K}\|_F^2. \tag{14}$$

Thus, we have $\|K^* - \hat{K}\|_F \leq \|K^* - K^0\|_F$, which provides an improved estimate $\hat{K}$ to $K^*$.

$\square$

# B Formula Derivations of Kernel Self-expressive Learning

## B.1 Formula Derivations of Kernel Self-expressive Learning with Proximal p-norm

Due to the non-convex nature of the $\|Z\|_p^p$ term and the penalty term, finding a closed-form solution for the $Z$-update is not straightforward. However, we can relax the matrix optimization to an element-wise optimization without loss of generality, thanks to the format of our chosen objective function. Below, we present the deriving process:

$$\mathcal{L} = \lambda\|Z\|_p^p + \gamma \sum_{i,j} \max(z_{ij} - 1, 0) + \mathrm{Tr}(U^\top(Z - C)) + \frac{1}{2\rho}\|Z - C\|_F^2$$

$$= \lambda \sum_{i,j}(z_{ij})^p + \gamma \sum_{i,j} \max(z_{ij} - 1, 0) + \sum_{i,j} \frac{1}{2\rho}(z_{ij} - c_{ij})^2 + u_{ij}(z_{ij} - c_{ij})$$

$$= \sum_{i,j}(\lambda z_{ij}^p + \gamma \mathbb{I}_{(z_{ij}>1)} \cdot (z_{ij} - 1) + \frac{1}{2\rho}(z_{ij} - c_{ij})^2 + u_{ij}(z_{ij} - c_{ij})),$$

where $\mathbb{I}_{(z_{ij}>1)} = \begin{cases} 1, & \text{if } z_{ij} > 1, \\ 0, & \text{otherwise.} \end{cases}$ is an indicator function and $\mathcal{L}_p = \mathcal{L} + \|\phi(X) - \phi(X)C\|_F^2$.

We can use a solver to find out the $Z$-update via

$$\frac{\partial \mathcal{L}_p}{\partial z_{ij}} = \lambda p z_{ij}^{p-1} + \gamma \mathbb{I}_{(z_{ij}>1)} + \frac{1}{\rho}(z_{ij} - c_{ij}) + u_{ij} = 0.$$

Using the Alternating Direction Method of Multipliers (ADMM) approach [61], we can update the variables as follows.

- $C$-update: Update $C$ by minimizing the augmented Lagrangian with respect to $C$:

$$C_{t+1} = \arg\min_C \mathcal{L}_p(C, Z_t, U_t)$$

$$= (2K + \frac{1}{\rho}I_n)^{-1}(2K + U_t + \frac{1}{\rho}Z_t),$$

  where $K := \phi(X)^\top \phi(X)$ and $I_n$ is the identity matrix of size $n \times n$.

- $Z$-update: Update $Z$ via solving

$$\frac{\partial \mathcal{L}_p}{\partial z_{ij}} = \lambda p z_{ij}^{p-1} + \gamma \mathbb{I}_{(z_{ij}>1)} + \frac{1}{\rho}(z_{ij} - c_{ij}) + u_{ij} = 0.$$

- $U$-update: Update $U$ via

$$U_{t+1} = U_t + \rho(Z_{t+1} - C_{t+1}).$$

We employ the ADMM process to solve the problem using the augmented Lagrangian function. However, despite the existence of a derivation method for this process, the computation proves to be excessively slow due to the following factors:

- Matrix inversion: Inverting $(2K + \frac{1}{\rho}I_n)^{-1}$ has a computational complexity of $O(n^3)$.

- Solver for $Z$-update: The golden-section solver is chosen to update $z_{ij}$, resulting in an approximate computational complexity of $\log(\frac{1}{\epsilon})$ for each entry [73], which is relatively high.

Additionally, some potential complex solutions may arise when performing the element-wise update. As a result, we provide the algorithm for future researchers to further investigate and optimize.

## B.2 Formula Derivations of Kernel Self-expressive Learning with Schatten p-norm

For $\frac{1}{2} < p < 1$, we can define a relaxation of the optimization problem as:

$$\mathcal{L}_{S_p} = \|\phi(X) - \phi(X)UV^\top\|_F^2 + \frac{\lambda}{2}\|U\|_F^2 + \frac{\lambda}{2}\|V\|_F^2.$$

With this definition, we can employ gradient descent to update the variables iteratively. The main steps of the algorithm are as follows:

- $U$-update: Compute the gradient of $\mathcal{L}_{S_p}$ with respect to $U$:

$$\frac{\partial \mathcal{L}_{S_p}}{\partial U} = 2K(UV^\top - I)V + \lambda U,$$

  where $K := \phi(X)^\top \phi(X)$. Update $U$ according to the gradient:

$$U_{\text{new}} = U_{\text{old}} - \alpha \nabla_U \mathcal{L}_{S_p},$$

  where $\alpha$ is the learning rate.
- $V$-update: Use the closed-form solution

$$V = 2KU(2U^\top KU + \lambda I)^{-1}.$$

- Convergence condition:
  If the difference between the updated $U$ and the old $U$ is less than a pre-defined tolerance ($\|U - U_{\text{old}}\|_F^2 < tol$), the algorithm is considered as converged, and we can terminate the loop. Otherwise, continue iterating with the new $U$ and $V$ values.

Upon convergence, the affinity matrix can be computed as $A = (|C| + |C|^\top)/2$, where $C = UV^\top$. This algorithm provides an efficient way to update the Schatten $p$-norm in the proposed KSL-Sp model, enhancing its robustness and clustering accuracy.

## B.3 Formula Derivations of Adaptive Kernel Least-Squares Representation

The problem that we are doing with is:

$$\min_{K \succeq 0, C} \|K - K^0\|_F^2 + \text{Tr}(K - 2KC + C^\top KC) + \lambda\|C\|_F^2.$$

Then we derive the augmented Lagrange equation in preparation for ADMM.

- By introducing an auxiliary variable A, we have:

$$\min_{K \succeq 0, A, C} \|K - K^0\|_F^2 + \text{Tr}(A - 2AC + C^\top AC) + \lambda\|C\|_F^2$$
$$\text{s.t. } K = A.$$

- Augmented Lagrange equation is calculated by

$$\mathcal{L}(K, A, C, U) = \|K - K^0\|_F^2 + \text{Tr}(A - 2AC + C^\top AC) + \lambda\|C\|_F^2$$
$$+ \text{Tr}(U^\top(K - A)) + \frac{\rho}{2}\|K - A\|_F^2,$$

  where $U$ is Lagrange multiplier, $\rho$ is the updating step size.

ADMM consists of iteratively updating $K$, $A$, $C$, and $U$ as follows:

1. $K$-update: $K = \arg\min_K \mathcal{L}(K, A_t, C_t, U_t)$. By First-Order Necessary Condition (FONC), we have

$$\frac{\partial \mathcal{L}}{\partial K} = 0 \Rightarrow K = \frac{2K^0 - U + \rho A}{\rho + 2}.$$

2. $K$-PSD: $K_{t+1} \leftarrow \mathcal{P}_1(K)$ by projecting $K$ onto the PSD set via Eq. (6).

3. $A$-update: $A_{t+1} = \arg\min_A \mathcal{L}(K_{t+1}, A, C_t, U_t)$. By FONC, we have

$$\frac{\partial \mathcal{L}}{\partial A} = \rho A + (I - 2C^\top + CC^\top - U - \rho K) = 0$$
$$\Rightarrow A = \frac{\rho K + U + 2C^\top - CC^\top - I}{\rho}.$$

4. $C$-update: $C_{t+1} = \arg\min_C \mathcal{L}(K_{t+1}, A_{t+1}, C, U_t)$. By FONC, we have

$$\frac{\partial \mathcal{L}}{\partial C} = (2\lambda I + A + A^\top)C - 2A^\top = 0$$
$$\Rightarrow C = 2(2\lambda I + A + A^\top)^{-1}A^\top.$$

5. $U$-update:

$$U_{t+1} = U_t + \rho(K_{t+1} - A_{t+1}).$$

## C  Experimental Settings

### C.1  Datasets

To evaluate the performance of our proposed approaches, we select four benchmark datasets commonly used in machine learning and computer vision, including:

- **Yale64** [64] [3]: The Yale64 dataset consists of face images of 15 different individuals, each with different facial expressions, lighting conditions, and facial details. It contains a total of 165 images of size $64 \times 64$ pixels in 4,096-dimensional vectors.

- **Umist** [65] [4]: The Umist dataset consists of face images of 20 different individuals, each with different facial expressions, lighting conditions, and facial details. It contains a total of 575 images of size $32 \times 32$ pixels in 1,024-dimensional vectors.

- **USPS** [66] [5]: The USPS dataset consists of handwritten digit images of size $16 \times 16$ pixels in 10 classes. For the five repeated experiments in this paper, we use five different random seeds to randomly select 1,000 images with 256-dimensional vectors as a subset.

- **Isolet** [67] [6]: The Isolet dataset is a speech dataset that contains recordings of 26 different speakers pronouncing the names of the English alphabet. It contains 1,560 recordings and each recording is of 617-dimensional vectors.

### C.2  Baselines

The proposed approaches are evaluated against a range of baseline methods, including

- **ZERO**: This statistical imputation method replaces missing values with zeros.

- **MEAN**: This statistical imputation method replaces missing values with the mean value of all observed values in the corresponding features.

- **$k$NN** [26]: This statistical imputation method predicts missing values based on the values of the $k$-nearest neighbors in the available data. (default value $k = 10$)

- **EM** [68]: This statistical imputation method estimates the missing values based on the Expectation Maximization algorithm.

- **SVT** [27]: This matrix completion method uses a singular value thresholding algorithm to find the low-rank matrix that best fits the available data.

- **GR** [28]: This matrix completion method uses an online algorithm to estimate the low-rank subspace of the data and then updates the subspace with new data points.

---

[3]https://www.kaggle.com/datasets/olgabelitskaya/yale-face-database
[4]https://cs.nyu.edu/ roweis/data/umist_cropped.mat
[5]https://cs.nyu.edu/ roweis/data/usps_all.mat
[6]http://archive.ics.uci.edu/ml/datasets/ISOLET

- **KFMC** [29]: This matrix completion method uses a kernelized factorization approach to estimate the high-rank matrix that best fits the available data. (default: polynomial kernel)
- **DC** [38]: This distance calibration method calibrates an initial distance matrix through double centering algorithm and transformation between similarity and distance.
- **TRF** [35]: This distance calibration method calibrates an initial distance matrix to satisfy all the triangle inequalities by solving a convex optimization problem.
- **EE** [39]: This distance calibration method calibrates an initial distance matrix to a Euclidean embeddable matrix by solving a convex optimization problem. (default value $\gamma = \frac{0.02}{\max\{d_{ij}^0\}}$)

### C.3 Implementation Details

**Data Preprocessing.** To obtain better clustering performance, we conduct different preprocessing strategies. In the Yale64 and USPS datasets, we use the original data without preprocessing; in the Umist and Isolet datasets, we perform the normalization to normalize all data samples into $[-1, 1]$.

**Generation of Incomplete Data.** For experiments in Sections 5.1, 5.2 and 5.3, we generate an incomplete data matrix $X^0$ by randomly replacing each value in the data matrix $X \in \mathbb{R}^{d \times n}$ with an NA value with a given probability $r$ (missing completely at random). We repeat this process five times and report the average performance of the baseline methods using $X^0$ as input.

**Data Imputation Methods.** We first impute the missing values in $X^0$ and then calculate the kernel matrix based on the imputed matrix $\hat{X}$. We then perform spectral clustering on the kernel matrix.

**Distance Calibration Methods.** We first estimate an initial distance matrix $D^0$ on the incomplete data matrix $X^0$ via Eq. (1). We then calibrate $D^0$ to $\hat{D}$ and calculate the kernel matrix based on $\hat{D}$. We finally perform spectral clustering on the resulting kernel matrix.

## D  Validation of Kernel Correction on Kernel and Distance Estimation

Two errors, relative-mean-square error (**RMSE** $= \frac{\|\hat{A} - A^*\|_F^2}{\|A^0 - A^*\|_F^2}$) and relative error (**RE** $= \frac{\|\hat{A} - A^*\|_F}{\|A^*\|_F}$), are used to evaluate the estimation accuracy, where $A^0$ is the naive kernel/distance matrix obtained by Eq. (1), $\hat{A}$ is the estimated one using different methods, and $A^*$ as the ground-truth. Besides, **Recall** measures the search accuracy of local relationships, which is defined as the overlapping ratio of two sets of top-10 similar items obtained by $\hat{A}$ and $A^*$, respectively.

Table 5: Quality comparisons of Gaussian kernel and Euclidean distance under a missing ratio $80\%$.

| Dataset | Metric | Naive | ZERO | MEAN | $k$NN | EM | SVT | GR | KFMC | DC | TRF | EE | KC |
|---|---|---|---|---|---|---|---|---|---|---|---|---|---|
| Yale64 | RMSE-K↓ | 1.000 | 11.52 | 3.009 | 11.46 | 3.009 | 11.39 | 11.13 | 8.838 | 2.564 | 0.994 | 0.741 | **0.624** |
| | RMSE-D↓ | 1.000 | 12.45 | 76.82 | 14.26 | 76.82 | 12.52 | 12.27 | 11.62 | 37.97 | 0.995 | 0.755 | **0.674** |
| | RE-K↓ | 0.113 | 0.382 | 0.195 | 0.381 | 0.195 | 0.380 | 0.376 | 0.335 | 0.180 | 0.112 | 0.097 | **0.089** |
| | RE-D↓ | 0.064 | 0.227 | 0.564 | 0.243 | 0.564 | 0.228 | 0.225 | 0.219 | 0.396 | 0.064 | 0.056 | **0.053** |
| | Recall↑ | 0.721 | 0.063 | 0.275 | 0.063 | 0.275 | 0.066 | 0.070 | 0.183 | 0.571 | 0.722 | 0.751 | **0.767** |
| Umist | RMSE-K↓ | 1.000 | 2.849 | 1.535 | 2.451 | 1.527 | 2.849 | 1.797 | 3.818 | 1.748 | 0.923 | 0.438 | **0.382** |
| | RMSE-D↓ | 1.000 | 20.80 | 27.64 | 21.20 | 27.62 | 20.80 | 8.397 | 14.05 | 110.4 | 0.916 | 0.440 | **0.431** |
| | RE-K↓ | 0.189 | 0.319 | 0.234 | 0.295 | 0.233 | 0.319 | 0.253 | 0.369 | 0.250 | 0.181 | 0.125 | **0.117** |
| | RE-D↓ | 0.107 | 0.487 | 0.561 | 0.492 | 0.561 | 0.487 | 0.309 | 0.400 | 1.122 | 0.102 | 0.071 | **0.070** |
| | Recall↑ | 0.726 | 0.092 | 0.171 | 0.119 | 0.172 | 0.092 | 0.596 | 0.248 | 0.226 | 0.740 | 0.771 | **0.785** |
| USPS | RMSE-K↓ | 1.000 | 0.471 | 0.292 | 0.411 | 0.393 | 0.469 | 0.389 | 0.475 | 0.386 | 0.547 | 0.262 | **0.223** |
| | RMSE-D↓ | 1.000 | 2.820 | 4.378 | 2.936 | 2.850 | 2.825 | 2.003 | 2.100 | 67.82 | 0.501 | 0.252 | **0.240** |
| | RE-K↓ | 0.460 | 0.316 | 0.248 | 0.295 | 0.287 | 0.315 | 0.287 | 0.317 | 0.286 | 0.340 | 0.235 | **0.217** |
| | RE-D↓ | 0.268 | 0.451 | 0.562 | 0.460 | 0.445 | 0.451 | 0.380 | 0.389 | 2.210 | 0.190 | 0.135 | **0.132** |
| | Recall↑ | 0.071 | 0.054 | 0.065 | 0.069 | 0.036 | 0.054 | 0.122 | 0.196 | 0.056 | 0.161 | 0.186 | **0.197** |
| Isolet | RMSE-K↓ | 1.000 | 1.188 | 0.796 | 0.966 | 0.993 | 1.188 | 1.001 | 1.390 | 0.956 | 0.817 | 0.339 | **0.291** |
| | RMSE-D↓ | 1.000 | 7.210 | 12.35 | 7.620 | 9.538 | 7.210 | 4.595 | 4.365 | 153.5 | 0.809 | 0.332 | **0.309** |
| | RE-K↓ | 0.282 | 0.307 | 0.251 | 0.277 | 0.280 | 0.307 | 0.282 | 0.332 | 0.275 | 0.255 | 0.164 | **0.152** |
| | RE-D↓ | 0.160 | 0.429 | 0.561 | 0.441 | 0.490 | 0.429 | 0.342 | 0.334 | 1.978 | 0.144 | 0.092 | **0.089** |
| | Recall↑ | 0.254 | 0.045 | 0.050 | 0.059 | 0.034 | 0.045 | 0.187 | 0.225 | 0.041 | 0.299 | 0.270 | **0.300** |

# E    Comprehensive Experiments on Standard Spectral Clustering

The robustness analysis is depicted in Fig. 1 in the main text, with accompanying numerical results in Table 6. Further, Fig. 2 displays clustering performance using ACC, NMI, and PUR metrics, underscoring the KC method's superiority over imputation baselines, particularly at large missing ratios. Notably, as the missing ratio escalates, imputation methods' accuracy and clustering efficacy decline sharply. Specifically, at missing ratios above 50%, their reliability wanes due to the paucity of data for imputation. In contrast, our KC method maintains consistent, and often superior, performance even with large missingness, exemplified by the results on the Yale64 and Umist datasets.

Table 6: Robustness analysis of kernel and distance estimation, neighborhood relationship, and standard spectral clustering on Umist. **Bold** font indicates best, and underline indicates second-best.

| Metric | Missing | Naive | ZERO | MEAN | $k$NN | EM | SVT | GR | KFMC | DC | TRF | EE | KC |
|---|---|---|---|---|---|---|---|---|---|---|---|---|---|
| RE-K↓ | 20% | 0.028 | 0.116 | 0.072 | **0.023** | 0.108 | 0.106 | 0.043 | 0.025 | 0.072 | 0.028 | 0.029 | 0.028 |
| | 50% | 0.066 | 0.232 | 0.157 | 0.115 | 0.180 | 0.232 | 0.111 | 0.076 | 0.162 | 0.065 | 0.062 | **0.057** |
| | 80% | 0.189 | 0.319 | 0.234 | 0.295 | 0.233 | 0.319 | 0.253 | 0.369 | 0.250 | 0.181 | 0.125 | **0.117** |
| RE-D↓ | 20% | **0.016** | 0.087 | 0.112 | 0.055 | 0.138 | 0.088 | 0.055 | 0.029 | 0.110 | **0.016** | 0.017 | **0.016** |
| | 50% | 0.037 | 0.237 | 0.302 | 0.222 | 0.281 | 0.237 | 0.144 | 0.090 | 0.372 | 0.037 | 0.035 | **0.033** |
| | 80% | 0.107 | 0.487 | 0.561 | 0.492 | 0.561 | 0.487 | 0.309 | 0.400 | 1.122 | 0.102 | 0.071 | **0.070** |
| Recall↑ | 20% | 0.954 | 0.887 | 0.912 | 0.952 | 0.870 | 0.887 | 0.944 | **0.964** | 0.932 | 0.954 | 0.950 | 0.956 |
| | 50% | 0.899 | 0.558 | 0.672 | 0.778 | 0.630 | 0.558 | 0.856 | 0.910 | 0.783 | 0.899 | 0.907 | **0.914** |
| | 80% | 0.726 | 0.092 | 0.171 | 0.119 | 0.172 | 0.092 | 0.596 | 0.248 | 0.226 | 0.740 | 0.771 | **0.785** |
| ARI↑ | 20% | 0.455 | 0.370 | 0.405 | 0.451 | 0.388 | 0.380 | 0.436 | 0.439 | 0.427 | 0.456 | **0.466** | 0.460 |
| | 50% | 0.443 | 0.258 | 0.332 | 0.326 | 0.325 | 0.256 | 0.398 | 0.448 | 0.361 | 0.434 | 0.450 | **0.451** |
| | 80% | 0.373 | 0.070 | 0.206 | 0.082 | 0.207 | 0.067 | 0.304 | 0.140 | 0.216 | 0.370 | 0.371 | **0.377** |

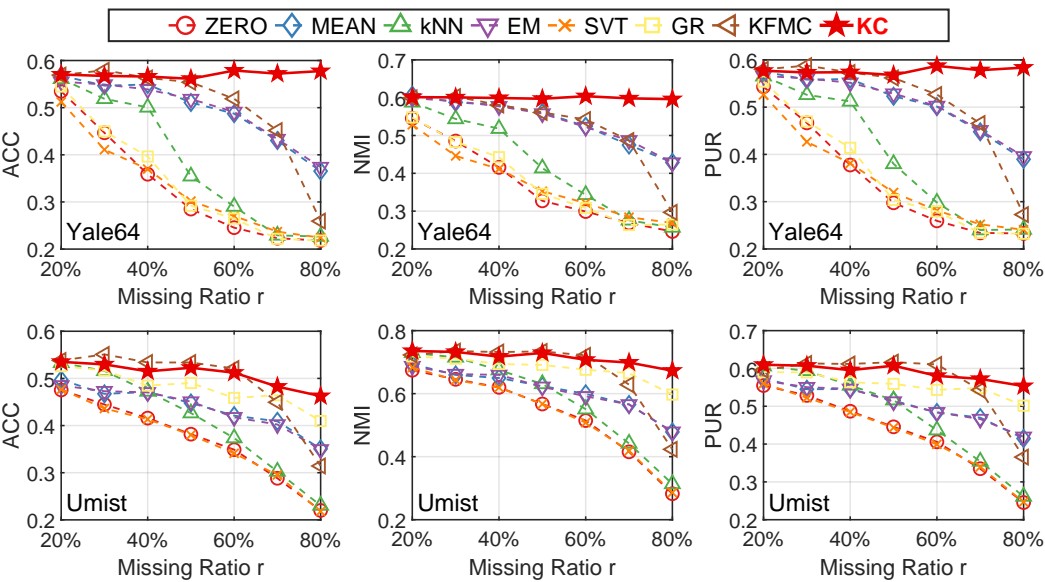

Figure 2: Robustness analysis of standard spectral clustering using the Gaussian kernel on the Yale64 and Umist datasets under a wide range of missing ratios, i.e., $r \in [20\%, 80\%]$.

# F    Comprehensive Experiments on Self-expressive Affinity Learning

**Hyperparameters of Affinity Learning Algorithms.**    All kernel self-expressive affinity learning algorithms in this study adhere to a unified framework, i.e., $\min_C \frac{1}{2}\|\phi(X) - \phi(X)C\|_F^2 + \lambda\mathcal{R}(C)$, where $\lambda$ is a regularization hyperparameter. Empirically, we choose $\lambda$ from the interval $[0, 30]$ to roughly achieve better clustering performance, as detailed in Table 7. Note that the optimal $\lambda$ may vary across algorithm implementations under different settings.

Table 7: Values of the hyperparameter $\lambda$ in all experiments in Section 5.3.

| Method | Yale64 | Umist | USPS | Isolet |
|--------|--------|-------|------|--------|
| KSSC | 0.08 | 0.1 | 0.07 | 0.06 |
| KLSR | 8 | 25 | 8 | 25 |
| KSL-Sp | 0.6 | 1 | 10 | 10 |
| AKLSR | 0.5 | 0.4 | 0.3 | 0.5 |

**Clustering Performance on the Exponential Kernel.** We investigate the benefits of the corrected Exponential kernel for kernel self-expressive affinity learning, including KSSC [19] and KLSR [3]. Tables 8 shows that our KC algorithm outperforms baselines regarding the NMI metric on the Exponential kernel based clustering, offering a systematic solution to clustering incomplete data.

Table 8: NMI Performance of self-expressive affinity on Exponential kernel with $80\%$ missing.

| Method | Dataset ($\lambda$) | Naive | ZERO | MEAN | $k$NN | EM | SVT | GR | KFMC | DC | TRF | EE | KC |
|--------|---------|-------|------|------|-----|------|------|------|------|------|------|------|------|
| KSSC | Yale64 $_{(0.03)}$ | 0.209 | 0.366 | 0.632 | 0.371 | 0.635 | 0.354 | 0.362 | 0.283 | 0.596 | 0.203 | 0.624 | **0.638** |
| | Umist $_{(0.03)}$ | 0.115 | 0.614 | 0.665 | 0.583 | 0.662 | 0.613 | 0.669 | 0.326 | 0.657 | 0.110 | 0.676 | **0.684** |
| | USPS $_{(0.05)}$ | 0.020 | 0.039 | 0.368 | 0.250 | 0.227 | 0.039 | 0.356 | 0.348 | 0.022 | 0.017 | **0.479** | 0.472 |
| | Isolet $_{(0.06)}$ | 0.064 | 0.069 | 0.123 | 0.085 | 0.094 | 0.069 | 0.120 | 0.132 | 0.071 | 0.060 | 0.634 | **0.661** |
| KLSR | Yale64 $_{(20)}$ | 0.614 | 0.304 | 0.573 | 0.309 | 0.565 | 0.295 | 0.312 | 0.328 | 0.586 | 0.618 | 0.625 | **0.634** |
| | Umist $_{(0.7)}$ | 0.127 | 0.570 | 0.655 | 0.557 | 0.653 | 0.566 | 0.682 | 0.487 | 0.642 | 0.130 | 0.688 | **0.693** |
| | USPS $_{(25)}$ | 0.083 | 0.309 | 0.433 | 0.354 | 0.217 | 0.310 | 0.341 | 0.381 | 0.440 | 0.217 | 0.513 | **0.515** |
| | Isolet $_{(20)}$ | 0.453 | 0.413 | 0.571 | 0.468 | 0.417 | 0.417 | 0.543 | 0.510 | 0.530 | 0.517 | 0.659 | **0.676** |

**Trade-off between Efficiency and Performance of the KC Algorithm.** To enhance the efficiency of the KC algorithm, we explore the use of randomized singular value decomposition (rSVD) [53] - a method that focuses on identifying the top-$k$ singular values (refer to Section 3.3). We compare the performance of the KC method with SD or rSVD on the USPS dataset. The results, presented in Table 9, reveal that rSVD greatly enhances the operational efficiency of the KC algorithm. However, it is crucial to choose an appropriate value for $k$ (number of top singular values), as it heavily impacts the quality of clustering. Smaller $k$ values (e.g., 10 or 20) result in poorer clustering due to the loss of important singular value information. On the other hand, selecting a suitable $k$ value (e.g., 50 or 100) significantly reduces running time while maintaining clustering performance comparable to SD, particularly for the standard spectral clustering algorithm (SC). Striking a trade-off between efficiency and performance remains an intriguing avenue for future research.

Table 9: Comparisons of rSVD and SD in the KC algorithm on the USPS with $80\%$ missing.

| Method | rSVD | | | | SD |
|--------|--------|--------|--------|---------|------|
| Metric | $k = 10$ | $k = 20$ | $k = 50$ | $k = 100$ | |
| RE-K | 0.695 | 0.419 | 0.262 | 0.291 | **0.217** |
| NMI-SC | 0.026 | 0.164 | 0.431 | 0.439 | **0.472** |
| NMI-KSSC | 0.021 | 0.019 | 0.018 | 0.019 | **0.360** |
| NMI-KLSR | 0.021 | 0.122 | 0.210 | 0.018 | **0.485** |
| Time (sec) | **1.96** | 2.12 | 3.10 | 5.76 | 72.97 |

**Comparisons of KSL-Sp and AKLSR.** AKLSR outpaces KSL-Sp in execution speed. Algorithm 4 delineates each AKLSR update step explicitly, ensuring quicker convergence, whereas KSL-Sp in Algorithm 3 relies on gradient descent for $U$ updates, leading to extended convergence times. Moreover, AKLSR's performance can vary with the presence or absence of a PSD constraint. Table 10 showcases the efficiency of AKLSR without PSD and its effectiveness with PSD on the Umist dataset.

Table 10: Comparisons of KSL-Sp and AKLSR on Umist using the Gaussian kernel obtained by KC.

| Metric | SC | KSL-Sp | AKLSR w/o PSD | AKLSR w/ PSD |
|--------|------|--------|---------------|--------------|
| ARI | 0.377 | 0.395 | 0.398 | **0.403** |
| Time (sec) | - | 12.57 | **0.04** | 0.23 |

