# OpenReview forum: "Boosting Spectral Clustering on Incomplete Data via Kernel Correction and Affinity Learning"
_NeurIPS.cc/2023/Conference — NeurIPS 2023 poster_

### Official Review · Reviewer_hJi1 · 2023-06-29

**Soundness:** 3 good
**Presentation:** 2 fair
**Contribution:** 3 good
**Rating:** 5
**Confidence:** 3

**Summary:**

This paper proposes an imputation-free framework with two novel approaches to improve spectral clustering on incomplete data. Firstly, the authors introduce a new kernel correction method that enhances the quality of the kernel matrix estimated on incomplete data with a theoretical guarantee, benefiting classical spectral clustering on pre-defined kernels. Secondly, they develop a new affinity learning method that equips the self-expressive framework with ℓp-norm to construct an intrinsic affinity matrix with adaptive extensions.




**Strengths:**

The originality of this paper is satisfying, which proposes an imputation-free framework with two novel approaches to improve spectral clustering on incomplete data and the significance is OK.

The quality and clarity of this paper are satisfying based on the clear presentation of the imputation-free framework with two novel approaches.

**Weaknesses:**

1. The authors propose an imputation-free framework with two novel approaches to improve spectral clustering on incomplete data. However, the advantages of the proposed method in dealing with incomplete data are not clearly stated based on Section 3, i.e., considering how to recover the missing data. Thus, the authors are expected to analyze the merits of the proposed method in dealing with the incomplete data.

2. In section 3.1, the authors give different methods to provide a calibrated distance matrix with benefits for distance-based kernels and are not a universal solution for dealing with incomplete data in spectral clustering tasks. However, these separated methods may make the novelty of this paper separated and not focused. I think the authors can better stating the revisiting distance calibration methods in this part.

3. The adopted datasets in the paper are almost with small scales in the experiments. The authors can add one or more multi-view datasets with large scales for validating the clustering performance.

4. The improvements of the proposed method compared with other methods are not significant in the experiments, i.e., the RE_K of KC is just 0.217 on USPS.

5. The authors can add one or more recent methods for comparison in the experiment, which make the comparson of experimental results more comprehensively.

**Questions:**

1. The authors propose an imputation-free framework with two novel approaches to improve spectral clustering on incomplete data. However, the advantages of the proposed method in dealing with incomplete data are not clearly stated based on Section 3, i.e., considering how to recover the missing data. Thus, the authors are expected to analyze the merits of the proposed method in dealing with the incomplete data.

2. In section 3.1, the authors give different methods to provide a calibrated distance matrix with benefits for distance-based kernels and are not a universal solution for dealing with incomplete data in spectral clustering tasks. However, these separated methods may make the novelty of this paper separated and not focused. I think the authors can better stating the revisiting distance calibration methods in this part.

3. The adopted datasets in the paper are almost with small scales in the experiments. The authors can add one or more multi-view datasets with large scales for validating the clustering performance.

4. The improvements of the proposed method compared with other methods are not significant in the experiments, i.e., the RE_K of KC is just 0.217 on USPS.

5. The authors can add one or more recent methods for comparison in the experiment, which make the comparson of experimental results more comprehensively.

**Limitations:**

1. The authors propose an imputation-free framework with two novel approaches to improve spectral clustering on incomplete data. However, the advantages of the proposed method in dealing with incomplete data are not clearly stated based on Section 3, i.e., considering how to recover the missing data. Thus, the authors are expected to analyze the merits of the proposed method in dealing with the incomplete data.

2. In section 3.1, the authors give different methods to provide a calibrated distance matrix with benefits for distance-based kernels and are not a universal solution for dealing with incomplete data in spectral clustering tasks. However, these separated methods may make the novelty of this paper separated and not focused. I think the authors can better stating the revisiting distance calibration methods in this part.

3. The adopted datasets in the paper are almost with small scales in the experiments. The authors can add one or more multi-view datasets with large scales for validating the clustering performance.

4. The improvements of the proposed method compared with other methods are not significant in the experiments, i.e., the RE_K of KC is just 0.217 on USPS.

5. The authors can add one or more recent methods for comparison in the experiment, which make the comparson of experimental results more comprehensively.

---

> ### Author Rebuttal · Authors · 2023-08-09
>
> **Response to Reviewer hJi1**
>
> Thank you very much for your positive feedback on the originality of our proposed method and its significance. We are delighted to hear that you found *our work satisfying in terms of quality and clarity*. Your comments are greatly appreciated and will help us further improve our work, especially for the presentation and evaluation.
>
> ---
>
> **Comment 1**: The advantages of the proposed method in dealing with incomplete data are not clearly stated based on Section 3, i.e., considering how to recover the missing data. Thus, the authors are expected to analyze the merits of the proposed method in dealing with the incomplete data.
>
> **Response 1**: Thanks for the suggestion. We do agree that including a discussion on the merits and challenges between imputation methods and imputation-free methods would be beneficial to the readers. We will try to add such a discussion.
>
> ---
>
> **Comment 2**: I think the authors can better stating the revisiting distance calibration methods in this part.
>
> **Response 2**: We appreciate your valuable feedback. It will be revised accordingly.
>
> ---
>
> **Comment 3**: The authors can add one or more multi-view datasets with large scales for validating the clustering performance.
>
> **Response 3**: Thanks for your suggestion. We will evaluate the work on larger datasets, including multi-view datasets. (Due to limited time, we were unable to finish such an evaluation in the rebuttal stage.) If any suitable multi-view datasets can be recommended, we will appreciate it.
>
> ---
>
> **Comment 4**: The improvements of the proposed method compared with other methods are not significant in the experiments, i.e., the RE_K of KC is just 0.217 on USPS.
>
> **Response 4**: We agree that the improvements under some settings are not that significant, while the improvements are consistent under all settings.
>
> ---
>
> **Comment 5**: The authors can add one or more recent methods for comparison in the experiment, which make the comparson of experimental results more comprehensively.
>
> **Response 5**: As you suggested, we included an additional comparison method, Polynomial Matrix Completion (PMC) [1], in our experiments. The results presented in Table 1 demonstrate that our KC method consistently outperforms the PMC method. Specifically, the KC method achieves better distance and kernel estimation with smaller relative errors (RE_D and RE_K), resulting in improved clustering performance with higher ACC, NMI, and PUR scores. These findings further validate the effectiveness of our KC method in comparison to recent imputation techniques.
>
> **Table 1**: Comparison of PMC and KC on distance estimation, kernel estimation and standard spectral clustering (SC) with Gaussian kernels for incomplete datasets under a missing ratio of 80%.
>
> | Dataset-Method | Yale64-PMC | Yale64-KC | Umist-PMC | Umist-KC | USPS-PMC | USPS-KC | Mfeat-PMC | Mfeat-KC |
> |-|-|-|-|-|-|-|-|-|
> |RE_D $\downarrow$|0.147|**0.053**|0.325|**0.070**|0.314|**0.132**|0.368|**0.095**|
> |RE_K $\downarrow$|0.152|**0.089**|0.457|**0.117**|0.304|**0.217**|0.360|**0.168**|
> |SC-ACC $\uparrow$|0.548|**0.578**|0.397|**0.463**|0.476|**0.523**|0.508|**0.761**|
> |SC-NMI $\uparrow$|0.574|**0.596**|0.508|**0.673**|0.466|**0.472**|0.469|**0.758**|
> |SC-PUR $\uparrow$|0.555|**0.584**|0.473|**0.553**|0.556|**0.609**|0.526|**0.804**|
>
> [1] Fan, J., et al. "Polynomial matrix completion for missing data imputation and transductive learning." AAAI, 2020.

---

> > ### Comment · Reviewer_hJi1 · 2023-08-17
> >
> > I appreciate the replies from the authors and keep my rating after reading the replies.

---

> > > ### Author Response · Authors · 2023-08-18
> > > **Thanks for Your Reply and Welcome Further Discussion**
> > >
> > > Thank you for taking the time to read the replies and we value your feedback. If you have any further questions or concerns, please feel free to let us know.

---

### Official Review · Reviewer_GE1q · 2023-07-04

**Soundness:** 3 good
**Presentation:** 3 good
**Contribution:** 3 good
**Rating:** 7
**Confidence:** 3

**Summary:**

This paper studies the spectral clustering problem when there is missing data. The paper proposes a new algorithm for correcting the computed kernel matrix by projecting the matrix to the nearest symmetric PSD matrix, and using this corrected kernel for clustering. The paper also combines the new kernel correction algorithm with affinity learning to develop a new technique for learning affinity matrices for incomplete data.

Experimental evaluations show that the newly developed techniques outperform alternative methods for handing incomplete data with respect to standard clustering metrics.

**Strengths:**

The newly proposed algorithm for kernel correction is novel and interesting. It has potential applications and could inspire future research directions. The algorithm is conceptually quite simple and captures the theoretical intuition quite naturally. The experimental results suggest that the new algorithm outperforms alternative methods.

**Weaknesses:**

The running time of the kernel correction algorithm is quite slow - it requires computing the spectral decomposition of the kernel matrix at every iteration. It is good that the authors discuss this limitation in the paper, although it is a weakness of the algorithm. Improving the running time could be a future research direction.

It would be interesting to compare experimentally the benefit that kernel correction brings over simply doing nothing to correct the missing data (or doing something very naive to construct the k-NN graph for example).

**Questions:**

Every iteration of the KC algorithm requires computing the spectral decomposition. Given that only the top k eigenvectors are needed for spectral clustering, could it be possible to speed up the algorithm by, for example, computing only k eigenvectors at each iteration?

What is the performance experimentally of performing spectral clustering on the naive k-NN graph constructed from the incomplete data? Given that this will be much faster than running KC, it would be interesting to see the trade-off between running time and accuracy.

**Limitations:**

The authors have adequately addressed the limitations of their work.

---

> ### Author Rebuttal · Authors · 2023-08-09
>
> **Response to Reviewer GE1q**
>
> Thanks for your positive feedback on the novelty of our proposed method and its potential applications for future research directions. Your comments are greatly appreciated and will help us further improve our work, especially for convincing empirical evaluation.
>
> ---
>
> **Question 1**: Given that only the top k eigenvectors are needed for spectral clustering, could it be possible to speed up the algorithm by, for example, computing only k eigenvectors at each iteration?
>
> **Answer 1**: We appreciate your insightful suggestion. To enhance the efficiency of the KC algorithm, we explored the use of randomized singular value decomposition (rSVD) [1] - a method that focuses on identifying the top-$k$ singular values (refer to Section 3.3). By replacing the spectral decomposition (SD) with rSVD, we were able to significantly reduce the time complexity of the algorithm from $O(n^3)$ to $O(n^2 \cdot \log(k) + 2n \cdot k^2)$ while ensuring accurate decomposition.
>
> We compared the performance of the KC method with SD or rSVD on two large datasets, USPS-1k and Mfeat-2K. The results, presented in Table 1(a) and 1(b) respectively, reveal that rSVD greatly enhances the operational efficiency of the KC algorithm. However, it is crucial to choose an appropriate value for $k$ (number of top singular values), as it heavily impacts the quality of clustering. Smaller $k$ values (e.g., 10 or 20) result in poorer clustering due to the loss of important singular value information. On the other hand, selecting a suitable $k$ value (e.g., 50 or 100) significantly reduces runtime while maintaining clustering performance comparable to SD, particularly for the standard spectral clustering algorithm (SC). Striking a trade-off between efficiency and performance remains an intriguing avenue for future research.
>
> **Table 1(a)**: Comparison of rSVD and SD on kernel correction and spectral clustering for the incomplete *USPS-1K* dataset under a missing ratio of 80%. Note that *RE_K* denotes the relative error of the corrected Gaussian kernel, *SC* denotes the standard spectral clustering algorithm, and *KLSR* denotes the kernel least-squares representation algorithm.
>
> |Method|rSVD (k=10)|rSVD (k=20)|rSVD (k=50)|rSVD (k=100)|SD|
> |-|-|-|-|-|-|
> |RE_K|0.695|0.419|0.262|0.291|**0.217**|
> |SC-ACC|0.153|0.290|0.494|0.509|**0.523**|
> |SC-NMI|0.026|0.164|0.431|0.439|**0.472**|
> |SC-PUR|0.189|0.332|0.572|0.582|**0.609**|
> |KLSR-ACC|0.147|0.266|0.317|0.148|**0.528**|
> |KLSR-NMI|0.021|0.122|0.210|0.018|**0.485**|
> |KLSR-PUR|0.173|0.296|0.389|0.176|**0.627**|
> |*Time (sec)*|**1.96**|2.12|3.10|5.76|72.97|
>
>
> **Table 1(b)**: Comparison of rSVD and SD on kernel correction and spectral clustering for the incomplete *Mfeat-2K* dataset under a missing ratio of 80%.
>
> |Method|rSVD (k=10)|rSVD (k=20)|rSVD (k=50)|rSVD (k=100)|SD|
> |-|-|-|-|-|-|
> |RE_K|0.681|0.474|0.223|0.201|**0.168**|
> |SC-ACC|0.142|0.201|0.715|0.721|**0.761**|
> |SC-NMI|0.017|0.071|0.667|0.733|**0.758**|
> |SC-PUR|0.148|0.213|0.748|0.777|**0.804**|
> |KLSR-ACC|0.136|0.178|0.629|0.606|**0.705**|
> |KLSR-NMI|0.012|0.045|0.563|0.613|**0.712**|
> |KLSR-PUR|0.140|0.184|0.675|0.686|**0.755**|
> |*Time (sec)*|**6.82**|7.28|9.08|13.98|390.84|
>
> [1] Halko N., et al. "Finding structure with randomness: Probabilistic algorithms for constructing approximate matrix decompositions." SIAM Review, 2011.
>
> ---
>
> **Question 2**: What is the performance experimentally of performing spectral clustering on the naive k-NN graph constructed from the incomplete data?
>
> **Answer 2**: As you suggested, we have compared the naive kNN graph and our KC method on four benchmark datasets. The results, presented in Table 2, show that the KC method surpasses the naive kNN graph across multiple aspects.
> - The KC method demonstrates improved accuracy in distance estimation and kernel estimation, as evidenced by smaller relative errors, i.e., RE_D and RE_K.
> - KC exhibits a better local relationship in the kNN graph, as indicated by a higher Recall value.
> - When applying SC, KSSC, and KLSR clustering algorithms, the KC method consistently outperforms the naive kNN graph in terms of ACC, NMI, and PUR metrics.
>
> Notably, while the naive kNN graph boasts a significantly faster runtime, its clustering performance is unstable and can be notably compromised when utilizing the KSSC or KLSR algorithms due to substantial errors in kernel estimation.
>
> **Table 2**: Comparison of the naive kNN graph and KC method on spectral clustering with standard Gaussian kernels (SC) and self-expressive affinity learning (KSSC, KLSR) for incomplete datasets under a missing ratio of 80%.
>
> | Dataset-Method | Yale64-Naive | Yale64-KC | Umist-Naive | Umist-KC | USPS-Naive | USPS-KC | Mfeat-Naive | Mfeat-KC |
> |-|-|-|-|-|-|-|-|-|
> | RE_D | 0.064 | **0.053** | 0.107 | **0.070** | 0.268 | **0.132** | 0.173 | **0.095** |
> | RE_K | 0.113 | **0.089** | 0.189 | **0.117** | 0.460 | **0.217** | 0.312 | **0.168** |
> | Recall | 0.721 | **0.767** | 0.726 | **0.785** | 0.071 | **0.197** | 0.246 | **0.286** |
> | SC-ACC | 0.561 | **0.578** | 0.462 | **0.463** | 0.343 | **0.523** | 0.728 | **0.761** |
> | SC-NMI | 0.588 | **0.596** | 0.669 | **0.673** | 0.222 | **0.472** | 0.740 | **0.758** |
> | SC-PUR | 0.572 | **0.584** | 0.549 | **0.553** | 0.395 | **0.609** | 0.782 | **0.804** |
> | KSSC-ACC | 0.190 | **0.586** | 0.119 | **0.496** | 0.152 | **0.427** | 0.121 | **0.713** |
> | KSSC-NMI | 0.219 | **0.616** | 0.101 | **0.714** | 0.018 | **0.360** | 0.010 | **0.648** |
> | KSSC-PUR | 0.198 | **0.601** | 0.131 | **0.584** | 0.171 | **0.529** | 0.124 | **0.750** |
> | KLSR-ACC| 0.582 | **0.607** | 0.485 | **0.488** | 0.144 | **0.528** | 0.205 | **0.705** |
> | KLSR-NMI| 0.606 | **0.616** | 0.676 | **0.696** | 0.019 | **0.485** | 0.078 | **0.712** |
> | KLSR-PUR| 0.592 | **0.613** | 0.568 | **0.572** | 0.176 | **0.627** | 0.217 | **0.755** |
> | *Time (sec)*| 0.18 | **0.03** | **0.54** | 8.05 | **0.60** | 72.97 | **3.76** | 390.84 |

---

> > ### Comment · Reviewer_GE1q · 2023-08-16
> >
> > Thank you for your response. I am pleased to see the additional experimental results based on computing fewer singular vectors, and a comparison with the naive algorithm. The trade-off between running time and performance in both experimental results is interesting and I feel is worth including in the next version of the paper.
> >
> > Based on this response, I am happy to increase my rating to 7.

---

> > > ### Author Response · Authors · 2023-08-17
> > > **Many Thanks to Your Great Support**
> > >
> > > Thank you for your valuable comments. We will include these additional results in the paper revision, as well as explore the trade-off between efficiency and performance in future research. Your feedback has been instrumental in enhancing the quality of our work, and we are grateful for your support.

---

### Official Review · Reviewer_7MjD · 2023-07-06

**Soundness:** 3 good
**Presentation:** 3 good
**Contribution:** 3 good
**Rating:** 7
**Confidence:** 4

**Summary:**

The authors introduce an imputation-free framework for correcting a kernel obtained from incomplete data. They propose the corrected kernel to be a PSD matrix with bounded elements that is closest to the initial kernel (calculated from incomplete data) in Frobenius norm. They show that the corrected kernel is guaranteed to be closer to the ground truth than the initial kernel. In the case of the Gaussian kernel, Dykstra's projection algorithm is presented to obtain the corrected kernel. They then extend the existing self-expressive affinity learning framework by incorporating the proximal p-norm (0 < p < 1) and the Schatten p-norm (½ < p < 1) penalties on the affinity matrix. An algorithm (KSL-Pp) based on augmented Lagrangian and ADMM is proposed for the proximal p-norm penalty. Another algorithm (KSL-Sp) based on gradient descent is proposed for the Schatten p-norm penalty. Finally, the authors combine the kernel correction and the self-expressive affinity learning frameworks to jointly learn the corrected kernel and the affinity matrix. An algorithm (AKLSR) based on augmented Lagrangian and ADDM is proposed to solve the joint optimization problem.

For numerical experiments, they perform a comparative analysis on synthetic and image datasets and show their proposed method produces a more accurate Gaussian kernel than the existing techniques. The performance of spectral clustering improves when their proposed corrected kernel is used against the corrected kernels obtained by competing techniques. For self expressive affinity learning, they show that more accurate affinity can be estimated when their proposed corrected kernel is used against the corrected kernels obtained by other techniques.

**Strengths:**

Significance: The Gaussian kernel plays a central role in several tasks including clustering, dimensionality reduction, graph neural networks etc. A comparative analysis on image datasets is provided that showed improved quality of the corrected kernel using the proposed approach against those obtained by existing techniques. The same is also reflected in the improved performance of spectral clustering and the improved quality of self-expressive affinity matrices based on their proposed corrected kernel. On a high level, these improvements could be helpful in downstream applications.

Quality: The authors adapted Dykstra's projection algorithm which comes with linear convergence guarantees, to correct the Gaussian kernel obtained from incomplete data.

**Weaknesses:**

Quality / clarity:
The authors extended the existing self-expressive affinity learning framework by incorporating p-norm based penalties (0 < p < 1). However, it is not clear how p-norm based penalties are more effective than the conventional 1-norm, Frobenius norm and nuclear norm based penalties. The provided empirical analysis seems insufficient to reach any conclusion.

The authors also proposed an algorithm that jointly optimizes the corrected kernel and the self-expressive affinity matrices using ADMM. However, there seems to be some issue in the formulation (see questions), and no empirical analysis is provided. This gives the sense that the paper is incomplete: the authors propose Adaptive Kernel Self-expressive Learning in 4.2 to tie between the kernel correction in section 3 and the self expressive affinity in 4.1, however do not include any simulations for this approach. Without demonstrating its efficacy, the inclusion of the proposed algorithm AKLSR seems unnecessary, and then the paper is essentially two unrelated approaches (kernel correction, self expressive learning) grouped together.

Missing references:
* Gilbert, A. C., & Jain, L. (2017, October). If it ain't broke, don't fix it: Sparse metric repair. In 2017 55th Annual Allerton Conference on Communication, Control, and Computing
* Shahid, N., Kalofolias, V., Bresson, X., Bronstein, M., & Vandergheynst, P. (2015). Robust principal component analysis on graphs. In Proceedings of the IEEE International Conference on Computer Vision
* Biswas, Arijit, and David W. Jacobs. "An Efficient Algorithm for Learning Distances that Obey the Triangle Inequality." BMVC. 2015.

**Questions:**

Major comments:
* Line 42. “Incomplete data” - the authors never define what incomplete setting they are solving, I assume missing at random, but this should be clarified.
* Line 128 “With a small missing ratio of features, $D_0$ already satisfies most triangle inequalities,..., then the algorithm typically yields only marginal improvement” - wouldn’t this be true of all methods? If the ratio of missing features is low, the impact on kernel construction is small and fixing it should yield marginal improvement.
* line 140, an explanation of “the quality of $\hat{D}$ cannot be guaranteed” would be helpful.
* Line 221 “the corrected distance obtained from the corrected Gaussian kernel will also be more accurate than the calibrated distance from the Euclidean embedding method.” is there a proof for this or just empirical evidence in the simulation?
* Why do the elements of \hat{C} on line 11 and 10 of algorithms 2 and 3 lie in [0,1]?
* The constraints on K in Eq. (9), seem incorrect. Why are they not consistent with those in Eq.(5)? Why is there no constraint that K is PSD?
* Line 201: O(n^2) storage limits the size of data that can be analyzed. Are the estimated kernels dense or sparse?
* Is the inverse of the formula for k_{ij} on line 260 used to obtain d_{ij}?
* The future work on line 287 “with future work on the potential of deep learning clustering techniques” should be made a bit more clear.

Simulations: My main concern is with the experiments.
1. The asymptotic time complexity provided by the authors is appreciated. It would be helpful if the actual time taken (in seconds) by the kernel correction and the affinity learning algorithms are also reported.
2. Numerical experiments for KLRR and KSL-Pp in Table 3 are missing.
3. Numerical experiments for KLRR, KSL-Sp and KSL-Pp in Table 3 and 4 in the supplementary material are missing.
4. Numerical experiments for the proposed AKLSR algorithm that jointly corrects the kernel and learns the affinity matrix are missing.
5. Gains are modest compared to EE - how do the two compare in terms of runtime/storage?
6. A popular measure to evaluate clustering is the adjusted Rand index (ARI). The authors should add this since it is more informative than ACC.
7. An additional non-image dataset would be useful to add if there is time (word-document, recommendation systems).
8. std values are missing from all the tables.
9. the experiments are performed only for 80% missing values. A wider range of missingness should be explored (plot can be used instead of tables) to demonstrate how robust the method is.
10. How are the hyperparameters of competing methods determined?

Minor comments:
* Although it seems intuitive, a reference for line 22 “incomplete data is commonly seen in practice, leading to inaccurate affinities and degraded clustering performance” would be helpful.
* A minus is missing in the formula for K_{ij} on line 87.
* The presentation in section 3 of “first method, “second method”,... seems odd. Are these the only relevant algorithms? Also, instead of using “first method” as a paragraph title, it would be better to use the name of the algorithm for the title of the paragraphs starting on lines 122, 130 and 142.
* A more precise reference (theorem number etc.) of Property 2 on line 146 would be helpful.
* Theorem 1 on line 193 may be placed after equation 5 as it holds in the general setting.
* The transition to section 4 is rather abrupt and 1-2 introductory sentences can help better tie the paper together.
* Which algorithm (SVD or rSVD) in Section 3.3 is used in the numerical experiments?
* What value of p for KSL-Sp used in Section 5.3?

**Limitations:**

limitations aren't discussed

---

> ### Author Rebuttal · Authors · 2023-08-10
>
> **Response to Reviewer 7MjD**
>
> Thanks for your valuable comments on our work. We greatly appreciate the time and effort you have dedicated to thoroughly evaluating our paper and providing detailed feedback. We will modify it accordingly.
>
> ---
>
> **Comment 1**: It is not clear how p-norm based penalties are more effective than the conventional 1-norm, Frobenius norm and nuclear norm based penalties.
>
> **Answer 1**: Theoretically, p-norm is a generalization of the L1 norm, offering a flexible sparsity control with a range of sparsity levels. Moreover, p-norm penalties exhibit robustness to outliers due to their smooth and continuous penalty term. Additional comparison results of different norms will be included in the revision to further validate these advantageous properties.
>
> ---
>
> **Comment 2**: Missing references.
>
> **Answer 2**: We will cite these important references accordingly.
>
> ---
>
> **Comment 3-1**: On the results of KSL-Pp and AKLSR algorithm.
>
> **Answer 3-1**: Sorry for any confusion caused by missing results. In practice, the KSL-Pp method requires handling numerous hyper-parameters and involves a non-convex optimization process during the Z-update step, making it difficult to effectively utilize. To address this limitation, we propose two extensions with the KSL-Sp and AKLSR algorithms. We haved conducted the experiments of AKLSR on two datasets as shown in Table 1, which partially validates the effectiveness of KC and AKLSR method. More results will be included in the revision.
>
> **Table 1(a)**: Performance of the AKLSR algorithm on Yale64 and Umist datasets under a missing ratio of 80%.
>
> |Dataset-Metric|ZERO|MEAN|kNN|EM|SVT|FNNM|GR|KFMC|DC|TRF|EE|KC|
> |-|-|-|-|-|-|-|-|-|-|-|-|-|
> |Yale64-NMI|0.332|0.610|0.334|0.602|0.312|0.614|0.341|0.307|0.564|0.385|0.577|**0.617**|
> |Yale64-PUR|0.300|0.591|0.312|0.581|0.281|0.602|0.307|0.290|0.546|0.348|0.555|**0.592**|
> |Yale64-ARI|0.071|0.377|0.079|0.373|0.057|0.382|0.074|0.051|0.321|0.116|0.329|**0.383**|
> |Umist-NMI|0.501|0.632|0.507|0.629|0.503|0.495|0.681|0.466|0.625|0.126|0.642|**0.687**|
> |Umist-PUR|0.428|0.519|0.419|0.521|0.425|0.418|0.567|0.424|0.525|0.145|0.540|**0.590**|
> |Umist-ARI|0.225|0.347|0.217|0.343|0.219|0.224|0.402|0.167|0.344|0.002|0.345|**0.398**|
>
> **Table 1(b)**: Comparison of AKLSR and SC algorithms on Yale64 and Umist datasets under a missing ratio of 80%.
>
> ||Yale64-SC|Yale64-AKLSR|Umist-SC|Umist-AKLSR|
> |-|-|-|-|-|
> |NMI|0.596|**0.617**|0.673|**0.687**|
> |PUR|0.584|**0.592**|0.553|**0.590**|
> |ARI|0.366|**0.383**|0.377|**0.398**|
>
> ---
>
> **Comment 3-2**: On comparison of EE and KC.
>
> **Answer 3-2**: While the KC method has comparable runtime and storage requirements, it offers the advantage of being applicable to a wide range of kernels. On the other hand, the EE method is limited to Laplacian kernels and relies on stricter assumptions, resulting in a narrower scope of applications.
>
> ---
>
> **Comment 3-3**: On the ARI metric, std values, and more missingness.
>
> **Answer 3-3**: Thanks for your suggestion and we will add these results in the revision.
>
> ---
>
> **Comment 3-4**: On experiments of a non-image dataset.
>
> **Answer 3-4**: We included an additional speech dataset, Isolet, in the Supplementary and also showed the superiority of our methods.
>
> ---
>
> **Comment 4-1**: Line 221. Is there a proof for this or just empirical evidence in the simulation?
>
> **Answer 4-1**: The claim is supported by empirical evidence from our experiments and is an intuitive observation. We will modify the claim more rigorously and seek to find a theoretical proof in future work.
>
> ---
>
> **Comment 4-2**: Why do the elements of \hat{C} on line 11 and 10 of algorithms 2 and 3 lie in [0,1]?
>
> **Answer 4-2**: Taking Algorithm 2 as an example, the constraint of $c_{ij} \in [0,1]$ in Eq. (6) has been incorporated through the term $\sum_{i,j} \max(z_{ij}-1,0)^2$ in the augmented Lagrangian function defined in Eq. (8). Consequently, when solving the augmented Lagrangian function using ADMM, the Z-update step largely guarantees the satisfaction of this constraint.
>
> ---
>
> **Comment 4-3**: The constraints on K in Eq. (9), seem incorrect. Why are they not consistent with those in Eq.(5)? Why is there no constraint that K is PSD?
>
> **Answer 4-3**: We apologize for any confusion caused by the inconsistency in Eq. (9). To construct a suitable Lagrangian function, we simplified the formulation and utilized a procedure that starts with an initial estimate $K^0$ without explicitly enforcing the PSD constraint. However, we recognize the need for a more rigorous investigation and a solution that incorporates the PSD constraint appropriately. We appreciate your feedback and will make sure to clarify this in our work.
>
> ---
>
> **Comment 4-4**: Line 201: O(n^2) storage limits the size of data that can be analyzed. Are the estimated kernels dense or sparse?
>
> **Answer 4-4**: Thanks for your suggestion. In practice, both the initial estimate $K^0$ and the corrected $\hat{K}$ are dense.
>
> ---
>
> **Comment 4-5**: Is the inverse of the formula for k_{ij} on line 260 used to obtain d_{ij}?
>
> **Answer 4-5**: Exactly. We obtain $d_{ij} = \sqrt{-\sigma^2 \log(k_{ij})}$ due to $k_{ij} = \exp(-d_{ij}^2/\sigma^2)$, where $\sigma = \text{median}\\{d_{ij}\\}$.
>
> ---
>
> **Comment 4-6**: Which algorithm (SVD or rSVD) in Section 3.3 is used in the numerical experiments?
>
> **Answer 4-6**: In our experiments, we used the spectral decomposition in the KC method. We will clarify it and include results of extensions with rSVD in the revision.
>
> ---
>
> **Comment 4-7**: What value of p for KSL-Sp used in Section 5.3?
>
> **Answer 4-7**: In practice, there is no need to explicitly specify the value of $p$ in the KSL-Sp method. Theoretically, we utilized the finding in previous work that for $\frac{1}{2} < p < 1$, $ ||C||_{S_p} = \min \frac{||U||_F^2 + ||V||_F^2}{2} $ with a constraint of $C = UV^{\top}$ holds true. Based on this, we constructed a relaxed Lagrangian function in Line 235 that does not involve the parameter $p$.

---

> > ### Comment · Reviewer_7MjD · 2023-08-18
> > **More clarification**
> >
> > I thank the authors for their replies.
> > Regarding the results in table 1(a) - AKLSR performs worse than KSL-Sp - what is the advantage then?
> > Are these averaged over multiple runs?
> >
> > I encourage the authors to include a result on a wider range of missingness values - it is hard to evaluate the performance based on a single value. it would be good to know how it performs in low/high missing rates

---

> > > ### Author Response · Authors · 2023-08-18
> > > **More Detailed Comparisons**
> > >
> > > Thanks for your feedback and suggestions. We acknowledge your willingness to participate in further discussions. The results in Table 1 are the average values for five runs. We will include these results in the revision.
> > >
> > > ---
> > >
> > > **Comparison of KSL-Sp and AKLSR:**
> > >
> > > When comparing KSL-Sp to AKLSR, we observe that AKLSR exhibits a significantly faster running speed. In Algorithm 4, each update step of AKLSR is expressed explicitly, leading to faster iterative convergence. However, KSL-Sp in Algorithm 3 requires solving through gradient descent to update $U$, resulting in longer convergence time.
> > >
> > >  Additionally, AKLSR can be further enhanced by incorporating a PSD constraint, resulting in AKLSR-PSD. Specifically, by introducing a spectral decomposition step after the $K$-update in Algorithm 4, we ensure that the kernel matrix $K$ used for updating the affinity matrix $C$ is PSD. The results presented in Table 2 provide partial validation of the efficiency of AKLSR and the effectiveness of AKLSR-PSD on the Umist dataset.
> > >
> > > **Table 2**: Comparison of SC, KSL-Sp, AKLSR, and AKLSR-PSD algorithms on the Umist dataset under a missing ratio 80%. We consider two different kernel matrices as inputs: the true Gaussian kernel matrix (TRUE) and the corrected kernel matrix (KC). We measure the time required to obtain a self-expressive affinity matrix from a given kernel matrix. All results are the average values for five runs.
> > >
> > >
> > > |Method|SC|KSL-Sp|AKLSR|AKLSR-PSD|
> > > |-|-|-|-|-|
> > > |TRUE-NMI|0.728|**0.778**|0.746|0.753|
> > > |TRUE-PUR|0.602|**0.674**|0.629|0.652|
> > > |TRUE-ARI|0.443|**0.563**|0.477|0.497|
> > > |TRUE-Time (sec)|-|12.574|**0.041**|0.254|
> > > |KC-NMI|0.673|**0.698**|0.687|0.691|
> > > |KC-PUR|0.553|0.592|0.590|**0.596**|
> > > |KC-ARI|0.377|0.395|0.398|**0.403**|
> > > |KC-Time (sec)|-|12.572|**0.039**|0.233|
> > >
> > > ---
> > >
> > > **More Results on Different Missingness:**
> > >
> > > As you suggested, we have compared the performance on the Umist dataset under a wider range of missingness. The results, presented in Table 3, highlight the superiority of the KC method over baselines, particularly for scenarios with a high missing ratio.
> > >
> > > - In terms of distance estimation and kernel estimation accuracy, the KC method consistently outperforms other methods, as indicated by smaller relative errors (RE_D and RE_K). Moreover, the KC method exhibits a stronger local relationship in the kNN graph, as evidenced by higher Recall values.
> > >
> > > - In cases where the missing ratio is small, the KC method has considerable performance or incremental improvement compard to imputation methods.
> > >
> > > - When dealing with a large missing ratio, imputation methods struggle due to the limited availability of observed data. This limitation increases the estimation error and leads to inaccurate kernel matrices, resulting in a significant degradation of clustering performance. In contrast, the KC method excels in terms of smaller errors and more stable clustering performance.
> > >
> > > **Table 3**: Performance of distance estimation, kernel estimation, and standard spectral clustering (SC) on the Umist dataset with a wide range of missing ratios, i.e., {20%, 50%, 80%}. Note that RE_D (RE_K) denotes the relative error of the corrected Euclidean distance (Gaussian kernel). All results are the average values for five runs.
> > >
> > > |Method|ZERO|MEAN|kNN|EM|SVT|FNNM|GR|KFMC|DC|TRF|EE|KC|rank|
> > > |-|-|-|-|-|-|-|-|-|-|-|-|-|-|
> > > |RE\_D-20\%|0.087|0.112|0.055|0.138|0.088|0.096|0.055|0.029|0.110|0.016|0.017|**0.016**|1|
> > > |RE\_D-50\%|0.237|0.302|0.222|0.281|0.237|0.254|0.144|0.090|0.372|0.037|0.035|**0.033**|1|
> > > |RE\_D-80\%|0.487|0.561|0.492|0.561|0.487|0.487|0.309|0.400|1.122|0.102|0.071|**0.070**|1|
> > > |RE\_K-20\%|0.116|0.072|0.023|0.108|0.106|0.068|0.043|0.025|0.072|0.028|0.029|**0.028**|2|
> > > |RE\_K-50\%|0.232|0.157|0.115|0.180|0.232|0.180|0.111|0.076|0.162|0.065|0.062|**0.057**|1|
> > > |RE\_K-80\%|0.319|0.234|0.295|0.233|0.319|0.319|0.253|0.369|0.250|0.181|0.125|**0.117**|1|
> > > |Recall-20\%|0.887|0.912|0.952|0.870|0.887|0.921|0.944|0.964|0.932|0.954|0.950|**0.956**|2|
> > > |Recall-50\%|0.558|0.672|0.778|0.630|0.558|0.680|0.856|0.910|0.783|0.899|0.907|**0.914**|1|
> > > |Recall-80\%|0.092|0.171|0.119|0.172|0.092|0.092|0.596|0.248|0.226|0.740|0.771|**0.785**|1|
> > > |SC-ARI-20\%|0.370|0.405|0.451|0.388|0.380|0.408|0.436|0.439|0.427|0.456|0.466|**0.460**|2|
> > > |SC-ARI-50\%|0.258|0.332|0.326|0.325|0.256|0.291|0.398|0.448|0.361|0.434|0.450|**0.451**|1|
> > > |SC-ARI-80\%|0.070|0.206|0.082|0.207|0.067|0.069|0.304|0.140|0.216|0.370|0.371|**0.377**|1|

---

> > > > ### Comment · Reviewer_7MjD · 2023-08-20
> > > > **Response**
> > > >
> > > > These additional experiments, as well as adding AKLSR-PSD, strengthen the contributions in my opinion. I am raising my score.

---

> > > > > ### Author Response · Authors · 2023-08-20
> > > > > **Many Thanks to Your Great Support**
> > > > >
> > > > > Thank you very much for your detailed and valuable comments. We are truly grateful for your positive feedback and raised score. Your suggestions and discussions will be carefully considered and incorporated into the revised version of our paper to further enhance its quality. Thank you once again for your support.

---

### Official Review · Reviewer_7HKb · 2023-07-16

**Soundness:** 3 good
**Presentation:** 3 good
**Contribution:** 2 fair
**Rating:** 5
**Confidence:** 3

**Summary:**

The paper proposes a new kernel correction method to address the issue of incomplete data. Existing approaches aim to recover the distance matrix of complete data, starting from that of the incomplete data. In contrast, the proposed method (Section 3.2) formulates the problem as finding a positive semi-definite matrix that is closest to kernel matrix of incomplete data. The optimisation is solved using a iterative approach.
The paper further extends the approach to (kernel) affinity learning problems to propose 3 further algorithms, and finally, experimentally shows that the proposed methods outperform existing distance completion methods in the setting of kernel spectral clustering.

**Strengths:**

- The proposed approach is technically sound, and based on a rather intuitively simple idea of projecting onto the space of positive semi-definite matrices.
- Extensions to affinity learning are proposed
- The experiments show clear improvement over existing approaches for completing distance matrices

**Weaknesses:**

Literature:
The paper focus only on kernel spectral clustering and affinity learning, and hence, misses the broad and older literature on kernel methods for supervised learning. A quick search of Google scholar reveals a considerable literature on this topic. Few papers are noted but the literature is quite large, and it is not clear why the paper does not compare with such approaches.
- Smola, Alex J., S. V. N. Vishwanathan, and Thomas Hofmann. "Kernel methods for missing variables." International Workshop on Artificial Intelligence and Statistics. PMLR, 2005.
- Dick, U., Haider, P., & Scheffer, T. (2008, July). Learning from incomplete data with infinite imputations. In Proceedings of the 25th international conference on Machine learning (pp. 232-239).
- Liu, X., Zhu, X., Li, M., Wang, L., Zhu, E., Liu, T., ... & Gao, W. (2019). Multiple kernel $ k $ k-means with incomplete kernels. IEEE transactions on pattern analysis and machine intelligence, 42(5), 1191-1204.
The paper needs to positioned well in context of this literature, and the proposed algorithm should be compared with existing kernel completion methods.

Weak theory, and insufficient experiments on different missing data:
Theorem 1 is rather weak since it only guarantees that if the incomplete kernel matrix was derived from a some true psd matrix, then the returned projection is closer to ground truth that the incomplete kernel matrix. However, the algorithm/theorem does not guarantee that the output kernel matrix is significantly better than the given incomplete one. A general guarantee cannot be provided without assumptions.
However, the work would significantly improve if there is a guarantee on recovery of ground truth kernel assuming that incomplete kernel matrix is obtained from data with features missing uniformly at random (setting considered in experiments)

On a similar note, the experiments only assume that features/values a missing at random. However, in practice, data is often systematically missing (some features tend to have higher missing values rate; there is often correlation between missing entries). Extensive experiments are needed to demonstrate that the proposed methods are robust to different types of missing entries, other than uniformly random.

**Questions:**

- The fundamental goal of the paper seems to be kernel completion. What is special about spectral clustering (or its affinity learning) in this context? Why cannot one use the methods for supervised learning?
- It is well known that spectral clustering does not require a positive semi-definite kernel (for instance, one can apply spectral clustering also on a graph adjacency). Then isn't it better to run any other matrix completion approach that does not impose psd?

**Limitations:**

The paper does not discuss any limitation, and the reported experimental results suggest that the proposed methods clearly outperform existing methods.
However, as noted in weakness, the paper needs comparison with existing kernel completion works and needs discussion on what kind of missing data can drastically impact the proposed methods.

Possible negative societal impact is neither evident nor discussed in the work.

---

> ### Author Rebuttal · Authors · 2023-08-10
>
> **Response to Reviewer 7HKb**
>
> Thanks very much for your detailed feedback on the contributions of our work. We are delighted to hear that you found *the proposed approach is technically sound* and *the experiments show clear improvement over existing approaches*. Your comments are greatly appreciated and will help us to further improve our work, especially for the presentation of the paper.
>
> Due to technical constraints, we are unable to upload the updated version of our paper at this point. Therefore, we will further polish it in the new version.
>
> ---
>
> **Comment 1**: Few papers are noted but the literature is quite large, and it is not clear why the paper does not compare with such approaches.
>
> **Response 1**: Thanks for your feedback. We do realize the literature in kernel learning with missing data under specific settings as in your recommended papers. Meanwhile, our work addresses **complete but inaccurate** (noisy) kernels due to the presence of incomplete observations, which is fundamentally different from most work in the literature primarily dealing with **incomplete** kernels.
>
> ---
>
> **Comment 2**: A general guarantee cannot be provided without assumptions. However, the work would significantly improve if there is a guarantee on recovery of ground truth kernel assuming that incomplete kernel matrix is obtained from data with features missing uniformly at random.
>
> **Response 2**: To clarify, Theorem 1 resides on a mild assumption that the true kernel matrix is PSD, which is typically assumed in previous studies. It asserts that if the initial kernel $K^0$ (complete but inaccurate) is not a PSD matrix, we can correct it to a closer estimate $\hat{K}$ to the unknown ground-truth $K^*$, providing a solid guarantee on the corrected kernel, i.e., $||K^*-\hat{K}||_F \le ||K^*-K^0||_F$. In fact, we can even provide a tighter performance bound for $\hat{K}$ with $||K^*-\hat{K}||_F \le 2||K^*-K^0||_2$, where the proof will be included in the revision.
>
> ---
>
> **Comment 3**: Extensive experiments are needed to demonstrate that the proposed methods are robust to different types of missing entries, other than uniformly random.
>
> **Response 3**: As you suggested, we have conducted additional experiments with different missing mechanisms, in addition to the setting of missing completely at random (MCAR) in our paper. For a given missing ratio, we generate a block of appropriate sizes located randomly in images and values in the block are missing where the missingness is systematically related to the location. Table 1 shows that our KC algorithm consistently outperforms existing data imputation and distance calibration methods under this block-missing mechansim. More results will be included in our further revision.
>
> **Table 1**: Performance of standard spectral clustering (SC) and self-expressive affinity learning (KSSC and KLSR) on the incomplete Yale64 dataset with the block-missing mechanism under a missing ratio of 80%.
>
> |Method|ZERO|MEAN|kNN|EM|SVT|FNNM|GR|KFMC|DC|EE|KC|
> |-----------------|-------|-------|-------|-------|-------|-------|-------|-------|-------|-------|-----------------|
> |SC-ACC|0.273|0.475|0.297|0.472|0.301|0.502|0.271|0.464|0.410|0.532|**0.535**|
> |SC-NMI|0.330|0.518|0.351|0.517|0.355|0.536|0.329|0.506|0.466|0.558|**0.562**|
> |SC-PUR|0.289|0.492|0.312|0.492|0.319|0.518|0.284|0.476|0.433|0.545|**0.547**|
> |KSSC-ACC|0.278|0.473|0.308|0.479|0.311|0.520|0.281|0.439|0.410|0.192|**0.533**|
> |KSSC-NMI|0.340|0.526|0.379|0.530|0.376|0.555|0.342|0.496|0.482|0.212|**0.562**|
> |KSSC-PUR|0.299|0.498|0.329|0.505|0.330|0.541|0.299|0.468|0.441|0.198|**0.542**|
> |KLSR-ACC|0.274|0.516|0.304|0.500|0.309|0.544|0.276|0.475|0.419|0.558|**0.568**|
> |KLSR-NMI|0.336|0.544|0.365|0.534|0.365|0.572|0.338|0.533|0.484|0.585|**0.592**|
> |KLSR-PUR|0.289|0.529|0.319|0.512|0.325|0.556|0.292|0.493|0.443|0.573|**0.579**|
>
> ---
>
> **Comment 4**: 1) The fundamental goal of the paper seems to be kernel completion. 2) What is special about spectral clustering (or its affinity learning) in this context? 3) Why cannot one use the methods for supervised learning?
>
> **Response 4**: **1)** Our work diverges from traditional kernel completion and aims at refining a complete yet inaccurate kernel matrix to yield a more precise estimate. **2)** The performance of spectral clustering algorithms relies heavily on the quality of the affinity matrix, often defined by kernels or self-expressive affinity. Our proposed algorithms are tailored to improve the affinity quality, rendering them particularly suitable for spectral clustering. **3)** Thanks for your suggestion. We agree the approach can be potentially applied to supervised learning tasks, such as nearest neighbor classification and information retrieval tasks. More work will be carried out along this line.
>
> ---
>
> **Comment 5**: 1) It is well known that spectral clustering does not require a positive semi-definite kernel. 2) Then isn't it better to run any other matrix completion approach that does not impose psd?
>
> **Response 5**: **1)** As you pointed out, spectral clustering does not necessitate the input to be a PSD kernel. Meanwhile in practice, spectral clustering often employs a Gaussian kernel, which is inherently PSD. Thus, our work adhered to the PSD constraints for the kernel, recognizing that other constraints may warrant further exploration. **2)** In our experimental evaluation, we conducted comparisons with four prevalent matrix completion methods for handling missing data, namely SVT, FNNM, GR, and KFMC. The results showed the superiority of our algorithm, affirming the validity of our approach within the context of spectral clustering.
>
> ---
>
> **Comment 6**: The paper does not discuss any limitation.
>
> **Response 6**: Thanks for the suggestion. We will include a discussion of limitation in the revision.

---

> > ### Comment · Reviewer_7HKb · 2023-08-10
> > **Continuing discussions on Responses 3 and 6**
> >
> > I thank the authors for the response, and for conducting new experiments. I would like to engage bit more on Comments 3,6.
> > - Please elaborate on Response 6, particularly what you see as potential limitations.
> > - The reason for my comment that the theory is weak is because the guarantee is $\Vert K^* - \hat{K}\Vert_F \leq \Vert K^* - K_0\Vert_F$, that is, in the worst case, the method need not give a better solution than $K_0$ (hence, its significance is not clear from a theoretical perspective). Hence, my comment was, under some assumptions, is the solution strictly better than $K-0$? More precisely, can one show  $\Vert K^* - \hat{K}\Vert_F \leq c\Vert K^* - K_0\Vert_F$ where $c\ll1$?

---

> > > ### Author Response · Authors · 2023-08-10
> > > **More Detailed Responses to Limitation and Theory**
> > >
> > > Thanks for your prompt response. We greatly appreciate your willingness to engage in further discussions and delve into the finer details.
> > >
> > > ---
> > >
> > > **Response to Limitation:**
> > >
> > > The potential limitations primarily stem from the time complexity of the method. The pre-iteration time complexity of the KC method is currently at $O(n^3)$, which poses challenges when dealing with large-scale datasets. To address this issue, a possible solution is to replace the spectral decomposition with a randomized singular value decomposition [1] (as mentioned in Section 3.3). This approach seeks top-$k$ singular values and effectively reduces the time complexity to $O(n^2 \cdot \log(k) + 2n \cdot k^2)$. However, the trade-off between efficiency and efficacy necessitates further investigation.
> > >
> > > [1] Halko N., et al. "Finding structure with randomness: Probabilistic algorithms for constructing approximate matrix decompositions." SIAM Review, 2011.
> > >
> > > ---
> > >
> > > **Response to Theory:**
> > >
> > > We have indeed taken into account the considerations you mentioned and conducted empirical results. Theoretically, if and only if $K^0$ is non-PSD, our estimate $\hat{K}$ is a better one in the sense that $||K^*-\hat{K}||_F < ||K^*-K^0||_F$.
> > >
> > > **1)** In the setting of missing completely at random, we generated a random matrix $X \in \mathbb{R}^{d \times n}$ with $x_{ij} \underset{\sim}{i.i.d.} \mathcal{N}(0,1)$ and presented the values of $\text{Prob}(K^0 \succeq 0|d,n,r)$ in **Figure 1. (d1, d2) in Appendix D.1**. Our observations are as follows:
> > >    + When $d/n$ is relatively small, the initial kernel matrix $K^0$ is non-PSD (with a high probability), resulting in $||K^*-\hat{K}||_F < ||K^*-K^0||_F$.
> > >    + When $(d,n)$ is given, a larger missing ratio $r$ would more likely result in a non-PSD $K^0$ so that $||K^*-\hat{K}||_F < ||K^*-K^0||_F$.
> > >
> > > **2)** The improvement from $K^0$ to $\hat{K}$ can be quantified by the relative-mean-square error ($\text{RMSE} = \frac{||K^*-\hat{K}||_F^2}{||K^*-K^0||_F^2}$), which shares similar results with the RMSE of Euclidean distance, as presented in our paper. Both **Figure 2 in the main text** and **Figure 2. (d1, d2) in Appendix D.2** showed that the RMSE of Euclidean distance can be significantlly smaller than 1 ($c\ll 1$), particularly for larger size $n$, larger missing ratio $r$, or smaller dimension $d$. This finding aligns with the RMSE results for kernels. For example, RMSE of kernel is about 0.22 in the USPS dataset with a missing ratio 80%. We will incorporate these into the revised version.
> > >
> > > **3)** Regarding upper bound for $||K^*-\hat{K}||_F$, we actually have built a tighter bound in the Response 2, i.e., $||K^*-\hat{K}||_F \le 2||K^*-K^0||_2$, where the spectral norm $||\cdot||_2$ is much smaller than the Frobenius norm $||\cdot||_F$ in practice, providing an improvement in the worst case. In fact, we have been considering this theoretical upper bound for a long time, but have not found a good solution because we do not make any distribution assumptions about $X$, such as low rank. We will continue to work on the theoretical bound of this method in the future.

---

> > > > ### Comment · Reviewer_7HKb · 2023-08-19
> > > > **Thanks for detailed response**
> > > >
> > > > I thank the authors for their detailed response on theory, and I see that they have also conducted detailed experiments to address other reviewers. I have increased my score

---

> > > > > ### Author Response · Authors · 2023-08-19
> > > > > **Many Thanks to Your Positive Feedback**
> > > > >
> > > > > Thank you for your time and valuable comments. We appreciate your positive feedback and increased score. We will incorporate your suggestions and discussions into the revision to further enhance our paper. Thanks again for your support.

---

### Decision · Program_Chairs · 2023-09-21

**Decision:**

Accept (poster)

**Comment:**

An interesting paper; which has seen an increase of the ratings of several reviewers (7HKb, 7MjD, GE1q) in the course of the discussion phase. Though the additional rebuttal and comments have increased the quality of the paper — assuming they make their way to the revision —, the authors could have done a better job of digging into some important comments; instead of mostly staying on the experimental side and maybe picking technical comments that did not require extensive digging. For example, I would have appreciated a thorough answer to comment 1 of hJi1.

The final impression is very much mitigated: the paper has undoubtedly scored some positive feedback at discussion the but it still stands pretty much on a borderline(+) global score.